


# Sources and mixing state of summertime background aerosol in the northwestern Mediterranean basin

Jovanna Arndt[1], Jean Sciare[2,3], Marc Mallet[4], Greg C. Roberts[4,5],
Nicolas Marchand[6], Karine Sartelet[7], Karine Sellegri[8], François Dulac[2], Robert
M. Healy[9], and John C. Wenger[1]

[1]Department of Chemistry and Environmental Research Institute, University College Cork, Cork,
Ireland
[2]LSCE, Laboratoire des Sciences du Climat et de l'Environnement, Unité Mixte
CEA-CNRS-UVSQ, Univ. ParisSaclay, CEA Saclay/Orme des Merisiers 701, F-91191
Gif-sur-Yvette, France
[3]Energy, Environment and Water Research Center, The Cyprus Institute, 2121 Nicosia, Cyprus
[4]CNRM, Centre National de Recherches Météorologiques UMR 3589, Météo-France/CNRS,
Toulouse, France
[5]Scripps Institution of Oceanography, Center for Atmospheric Sciences and Physical
Oceanography, La Jolla, United States
[6]Aix Marseille Univ, CNRS, LCE, Marseille, France
[7]CEREA, Centre d'Enseignement et de Recherche en Environnement Atmosphérique, Joint
Laboratory ENPC ParisTech/EDF R&D, Université Paris-Est, Marne la Vallée, France
[8]LaMP, Laboratoire de Météorologie Physique CNRS UMR6016, Observatoire de Physique du
Globe de Clermont-Ferrand, Université Blaise Pascal, Aubière, France
[9]Environmental Monitoring and Reporting Branch, Ontario Ministry of the Environment and
Climate Change, Toronto, Canada

*Correspondence to:* John C. Wenger (j.wenger@ucc.ie)

**Abstract.** An aerosol time-of-flight mass spectrometer (ATOFMS) was employed to provide real-time single particle mixing state and thereby source information for aerosols impacting the western Mediterranean basin during the ChArMEx-ADRIMED and SAF-MED campaigns in summer 2013. The ATOFMS measurements were made at a ground-based remote site on the northern tip

of Corsica Island. 27 distinct ATOFMS particle classes were identified and subsequently grouped into 8 general categories: EC-rich (elemental carbon), K-rich, Na-rich, Amines, OC-rich (organic carbon), V-rich, Fe-rich and Ca-rich. Mass concentrations were reconstructed for the ATOFMS particle classes and found to be in good agreement with other co-located quantitative measurements (PM$_1$, black carbon (BC), organic carbon, sulfate mass and ammonium mass). Total ATOFMS re-

constructed mass (PM$_{2.5}$) accounted for 70-90% of measured PM$_{10}$ mass and was comprised of regionally transported fossil fuel (EC-rich) and biomass burning (K-rich) particles. The accumulation of these transported particles was favoured by repeated and extended periods of air mass stagnation over the western Mediterranean during the sampling campaigns. The single particle mass spectra proved to be valuable source markers, allowing the identification of fossil fuel and biomass burn-

ing combustion sources, and therefore highly complementary to quantitative measurements made


by particle-into-liquid sampler ion chromatography (PILS-IC) and an aerosol chemical speciation monitor (ACSM), which have demonstrated that $PM_1$ and $PM_{10}$ were comprised predominantly of sulfate, ammonium and OC. Good temporal agreement was observed between ATOFMS EC-rich and K-rich particle mass concentrations and combined mass concentrations of BC, sulfate, ammonium

and low volatility oxygenated organic aerosol (LV-OOA). This combined information suggests that combustion of fossil fuels and biomass produced primary EC- and OC-containing particles, which then accumulated ammonium, sulfate and alkylamines during regional transport. Three other sources were also identified: local biomass burning, marine and shipping. Local combustion particles (emitted in Corsica) contributed little to $PM_{2.5}$ particle number and mass concentrations but were easily

distinguished from regional combustion particles. Marine emissions comprised fresh and aged sea salt; the former detected mostly during one 5-day event during which it accounted for 50-80% of sea salt aerosol mass, while the latter detected throughout the sampling period. Dust was not efficiently detected by the ATOFMS, and support measurements showed that it was mainly in the $PM_{2.5-10}$ fraction. Shipping particles, identified using markers for heavy fuel oil combustion, were associated

with regional emissions, and represented only a small fraction of $PM_{2.5}$ particle number and mass concentration at the site.

## 1   Introduction

The atmosphere in the Mediterranean basin is strongly influenced by numerous and varied aerosol sources. Anthropogenic emissions from heavily industrialised parts of Southern Europe (e.g. Genoa and Milan), the megacities of Istanbul and Cairo, a large range of smaller population centres dis-

seminated all over the basin, as well as intense shipping activities render the Mediterranean basin one of the most impacted zones on the planet to air pollution (de la Paz et al., 2013; Karanasiou et al., 2014). Natural sources such as Saharan dust, sea-spray and frequent forest fires exert further considerable stress on regional air quality (Kanakidou et al., 2011). Transport of air pollution

from outside the Mediterranean region is one cause for increased concentrations of primary and secondary pollutants (Lelieveld et al., 2002). In the summertime upper troposphere, Asian monsoon outflow transports pollution across northern Africa and the Mediterranean (Scheeren et al., 2003). In the middle troposphere, westerly winds prevail, transporting polluted air masses from western Europe and North America (Marmer and Langmann, 2005). In the surface layer, land emissions from

south and central Europe are transported to the eastern Mediterranean by northerly winds (Sciare et al., 2003). The geography and regional meteorological processes in the western Mediterranean also favour the accumulation and ageing of polluted air masses (Gangoiti, 2001; Lelieveld et al., 2002; Millán et al., 2000, 2002; Millán and Salvador, 1997; Rodríguez et al., 2002; Salvador et al., 1999; Soriano et al., 2001). Arid conditions, combined with high solar radiation and photochemical

conversion rates significantly enhance air pollution mostly in the form of $PM_{2.5}$ and $O_3$. The high-



est particulate matter (PM) concentrations are generally found in southern and eastern Europe and attributed to diverse emission sources such as industry, traffic, resuspended dust, shipping emissions and African dust intrusions (Karanasiou et al., 2007, 2009, 2011, 2014; Lelieveld et al., 2002; Querol et al., 2004; Rodríguez et al., 2007; Salameh et al., 2015). A number of studies have reported that in rural environments in the Mediterranean, airborne PM and ammonium sulfate concentrations undergo a seasonal cycle characterised by a summer maximum (Bergametti et al., 1989; Kubilay and Saydam, 1995; Querol et al., 1998a, b; Rodríguez et al., 2002). This seasonal cycle has not been reported at rural sites in central and northern Europe, where high PM events are mostly recorded in winter during stagnant episodes caused by cold temperature inversions and low wind speed (Beekmann et al., 2015; Favez et al., 2007; Monn et al., 1995; Röösli et al., 2001; Turnbull and Harrison, 2000). Long term measurements at sites in the western and eastern Mediterranean basins by Querol et al. (2009) showed that mineral matter is the major component of $PM_{10}$ (22-38%) in both areas, with relatively high proportions in $PM_{2.5}$ (8-14%), followed by sulfate, organic matter (OM), nitrate and ammonium. Most studies to date have documented Mediterranean aerosol properties in the eastern basin or at coastal continent-based sites in the western basin, where they were subject to the proximity of considerable urban or industrial emissions. Land-based measurements of the background composition of western Mediterranean atmospheric aerosol is best investigated on the shore line of relatively industry-free, less urbanised islands and studies in such locations have thus far been limited. One of the central aims of the ChArMEx (Chemistry-Aerosol Mediterranean Experiment; https://charmex.lsce.ipsl.fr) project is to make background aerosol observations in such islands as Corsica and the Balearic Islands. In addition, studies of the chemical composition of single aerosol particles in the Mediterranean are particularly scarce and are restricted to urban environments (Dall'Osto et al., 2013, 2016; Dall'Osto and Harrison, 2006; McGillicuddy, 2014). Single particle mass spectrometers, such as the aerosol time-of-flight mass spectrometer (ATOFMS), have proven valuable in identifying and characterising a wide variety of particle sources; sea salt, mineral dust, vehicle exhaust, tyre wear, solid fuel combustion (coal, peat and wood), shipping and various industrial emissions (Beddows et al., 2004; Bhave et al., 2001, 2002; Dall'Osto et al., 2014; Giorio et al., 2012; Harrison et al., 2012; Healy et al., 2009, 2010, 2012; Liu et al., 2003; Spencer et al., 2008; Tao et al., 2011). Mixing state information – both internal and external – provided by mass spectrometers has been used to determine the type of atmospheric processing particles have undergone (Gard et al., 1998), as well as their acidity and hygroscopicity (Denkenberger et al., 2007; Healy et al., 2015), properties which affect their ability to act as CCN (Furutani et al., 2008). Combination of ATOFMS and hygroscopicity tandem differential mobility analyser (HTDMA) data has shown that sea salt and particles containing amines and nitrate are hydrophilic, while the more organic carbon (OC) particles contain the more hydrophobic they are (Herich et al., 2009; Wang et al., 2014). Fresh particles can be distinguished from aged ones by the presence of secondary inorganic species such as nitrate, sulfate and ammonium (Cahill et al., 2012; Healy et al., 2010; Liu et al., 2003; Pratt and





Prather, 2009), which is also helpful in differentiating particles from local and transported sources (Healy et al., 2012). Single particle chemical speciation is therefore a useful tool, complementary to characterising particle optical and physical properties (Moffet and Prather, 2009), for examining the effect aerosols have on air quality and climate. Measurements from the ATOFMS also provide useful information to validate mixing-state resolved models (Zhu et al., 2016). In air quality and climate models, mixing-state information is essential as this property strongly impacts aerosol composition, hygroscopicity, optical and CCN properties over urban areas (Zhu et al., 2016). An important application of single-particle data is source apportionment of ambient aerosol. If a unique particle composition signature can be linked with a specific source, then number and mass concentration contributions can be estimated at a receptor site (Allen et al., 2000; Bein et al., 2006; Pratt and Prather, 2009; Qin et al., 2006; Reinard et al., 2007). Most chemical speciation measurements to date in the Mediterranean have focused on bulk aerosol, and while they provide quantitative data they do not resolve the mixing states of ambient particles. As such, source identification and therefore apportionment – often one of the main goals of aerosol measurements – is limited with these techniques. In this context, an aerosol time-of-flight mass spectrometer (ATOFMS) was employed to provide real-time single particle mixing state and thereby source information for aerosols impacting the western Mediterranean basin during two ChArMEx special observation periods in summer 2013: ADRIMED (Aerosol Direct Radiative Impact on the regional climate in the MEDiterranean region; Mallet et al., 2016) and SAF-MED (Secondary Aerosol Formation in the MEDiterranean).

## 2 Methodology

### 2.1 Sampling site and instrumentation

Measurements were performed at the atmospheric monitoring station in Ersa (coordinates: 42°58'09"N, 09°22'49"E), Cape Corsica, near the North tip of Corsica Island. This station is well positioned to investigate polluted air masses transported over the western Mediterranean basin from the highly industrialised regions of the Po Valley (Royer et al., 2010) and/or Marseille/Fos-Berre (El Haddad et al., 2011, 2013). The site was fully equipped for the measurement of aerosol chemical, physical and optical properties. This ground-based remote station is located at an altitude of 530 m above sea level (asl) at the remote Cape Corsica Peninsula and has unobstructed views to the sea over ~270°(Lambert et al., 2011). The ADRIMED and SAF-MED field campaigns took place from 11th June to 5th July and from 12th July to 6th August 2013, respectively, during the Mediterranean dry season over the western and central Mediterranean basins. Some of the key instruments deployed during the campaigns are given in Table 1. A full list of instruments deployed during ADRIMED can be found in the overview for this campaign by Mallet et al. (2016), while a summary of the main findings of the SAF-MED campaign is currently in preparation. The ATOFMS (TSI model 3800) was operated continuously from 12th June – 6th August, with a period of downtime from 12-18th July. A





detailed description of the ATOFMS can be found elsewhere (Gard et al., 1997). Briefly, it consists of (i) an aerodynamic focussing lens (TSI AFL100) Su et al. (2004) that transmits particles in the aerodynamic diameter ($D_a$) range 100-3000 nm, (ii) a particle sizing region, and (iii) a bipolar reflectron time-offlight mass spectrometer. Single particles are desorbed/ionized using a pulsed Nd:YAG laser ($\lambda$ = 266 nm, ~1 mJ pulse$^{-1}$). Positive and negative ion mass spectra of individual aerosol particles are obtained, which enable identification of the chemical constituents. Gradual degradation in the power of the sizing lasers during the campaign was observed and resulted in effective size ranges of 300-3000 nm $D_a$ for ADRIMED and 500-3000 nm $D_a$ for SAF-MED.

### 2.2 ATOFMS Data Analysis

Over 1.2 million single particle mass spectra were generated by the ATOFMS during the sampling period and clustered using the K-means algorithm ($K$=80), as described in detail elsewhere (Gross et al., 2010; Healy et al., 2009, 2010). Clusters exhibiting very similar average mass spectra (including those with the same major ions but varying relative signal intensities), comparable temporal trends and size distributions were merged. The final merged clusters were then identified as particle "classes"; 27 in total.

The particle class labelling scheme used herein is regularly used in the literature (Ault et al., 2010; Dall'Osto and Harrison, 2006; Pratt and Prather, 2012; Spencer and Prather, 2006) and indicates either the probable source (e.g. sea salt) or the dominant species in the positive ion mass spectra (e.g. K, EC, Fe etc.), with the order of the ions indicating their relative mass spectral intensities. For example, a particle class with high intensity mass spectral features for sodium and elemental carbon is labelled *Na-EC*. In some cases this is followed by a secondary species detected in the negative mass spectra (e.g. *K-NO$_X$*), which usually provides insight into the atmospheric ageing the particles have undergone locally or during transport (Reinard et al., 2007).

Single-particle mass spectrometers such as the ATOFMS do not provide quantitative information in the form of particle number or mass concentrations – rather the ATOFMS provides speciation in particle counts classified by aerodynamic diameter. The transmission biases of the AFL, the number of particles the system can size and ionise at any given time (data acquisition busy time), as well as limited detection of particles <150 nm (determined by wavelength of sizing lasers and the amount of scattered light) hinder full and accurate counting of particles over the entire ATOFMS size range (100-3000 nm $D_a$). The desorption/ionisation laser used by the ATOFMS also complicates quantitative speciation. Shot-to-shot fluctuations in laser output power and variations in power density (Gaussian) across the laser beam (Steele et al., 2005; Wenzel and Prather, 2004) create variance in the amount of a particle that is desorbed and can also lead to variations in resultant mass spectral peak height and area (Reinard and Johnston, 2008).

Given these limitations, in order to produce meaningful particle number and mass concentrations for ATOFMS particle classes the total ATOFMS counts were scaled using quantitative particle





counting instruments operated concurrently; an optical particle counter (OPC, TSI model 3300) and
160 a scanning mobility particle sizer (SMPS, TSI DMA model 3080 and CPC model 3010). Reconcil-
ing SMPS, OPS and ATOFMS data requires conversion of $D_m$ (electrical mobility diameter; SMPS)
and $D_o$ (optical diameter; OPC) measured with the SMPS and OPS into the corresponding ATOFMS
$D_a$ (aerodynamic diameter), using the following relationship:

$$D_a = \frac{\rho_p D_{ve}}{\rho_0 \chi}$$

Where $\rho_p$ is the particle density (discussed hereafter), $D_{ve}$ is the volume equivalent diameter (op-
erationally equivalent to $D_m$ or $D_o$), $\rho_0$ is standard density (1 g/cm$^{-3}$) and $\chi$ is the dynamic shape
factor (assumed to be 1, thus representing spherical shape). This is a simplified version of particle
diameter, morphology and density relationships that are covered in much greater detail elsewhere
(DeCarlo et al., 2004). No correction factors were used to merge the SMPS and OPS data; $D_m$ and
170 $D_o$ were both assumed to be equivalent to geometric diameter.

Knowledge of particle density is required for this conversion. This value can be estimated from
bulk mass concentration measurements made, for example, by an aerosol chemical speciation moni-
tor (ACSM) and an instrument which measures black carbon (BC), such as a multi-angle absorption
spectrometer (MAAP).

$$\textit{Average density} = \frac{m_{BC} + m_{org} + m_{Cl} + m_{SO_4} + m_{NO_3} + m_{NH_4}}{\frac{m_{BC}}{1.5} + \frac{m_{org}}{1.2} + \frac{m_{Cl}}{1.5} + \frac{m_{SO_4} + m_{NO_3} + m_{NH_4}}{1.5}}$$

### 2.3 Correlations

Reconstructed ATOFMS mass concentrations were then compared, using regression analysis and
the coefficient of determination, with those obtained by the TEOM (tapering element oscillating
microbalance; PM$_{10}$ and PM$_1$ mass concentrations), PILS-IC (PM$_{10}$ mass concentrations of SO$_4^{2-}$,
NH$_4^+$, NO$_3^-$, K$^+$, Ca$_2^+$, Na$^+$, Mg$^+$, Cl$^-$, oxalate, methanesulfonate – MSA), MAAP (BC mass con-
centrations) and ACSM (PM$_1$ mass concentrations of SO$_4^{2-}$, NH$_4^+$, NO$_3^-$, Cl$^-$, organic carbon).
Positive matrix factorisation (PMF) performed on the ACSM organic fragments produces factors
which correspond to a group of OA constituents with similar chemical composition and temporal
behaviour that are characteristic of different sources and/or atmospheric processes (Zhang et al.,
2011). Three factors were resolved; low-volatility oxygenated organic aerosol (LV-OOA, aged and
non-volatile), semi-volatile oxygenated organic aerosol (SV-OOA, less oxidised and aged than LV-
OOA) and hydrocarbon-like organic aerosol (HOA, representative of fossil fuel combustion). More
information on ACSM results and source apportionment are available in Michoud et al. (2016).

The coefficient of determination, as $R^2$, was used to evaluate the effectiveness of the ATOFMS
mass reconstruction. All $R^2$ values can be found in SI Table 1. This analysis was considered ap-
propriate as the objective of ATOFMS mass reconstruction was to produce mass concentrations as
similar as possible to other mass measurement techniques. $R^2$ values of <0.5, ~0.5-0.7 and >0.7 are



typically considered indicative of weak, moderate and strong linear (positive or negative) relationships respectively.

## 3 Results and Discussion

### 3.1 Air mass back trajectories

Back trajectory analysis was performed using the HYSPLIT model (Revision 631, July 2014) (Draxler and Hess, 1998) to identify air masses influencing the sampling site. The HYSPLIT model was run using the PC Windows based software available online (http://www.ready.noaa.gov/HYSPLIT.php) and meteorological input from the Global Data Assimilation System (GDAS) archive. 120-hour back trajectories ending 500 m above ground level (AGL) at Ersa (Corsica) were calculated for each hour between 12$^{th}$ June and 7$^{th}$ August 2013 (total: 1344 trajectories). Five broad periods with different air mass regimes were identified in the period based on the ATOFMS, OPS, ACSM, PM$_1$ and PM$_{10}$ temporal profiles. Separate cluster analyses were performed to classify the trajectories for each of these five campaign periods (Figure 1). A plot of total spatial variance as a function of the number of clusters was used to determine the number of clusters. The clustered HYSPLIT 120-hr back trajectories for each period were used to determine which air masses most influenced the ATOFMS measurements, and temporal profiles for particle numbers and mass concentrations have been labelled with these air mass origins (France, Mediterranean, Spain, North Atlantic, UK, Italy, eastern Europe).

Period 1 (12-20$^{th}$ June) was dominated by recirculating air masses over the Mediterranean Sea, Period 2 (20-26$^{th}$ June) by North Atlantic air masses, Period 3 (26$^{th}$ June-4$^{th}$ July) by trajectories passing largely over France, Period 4 (4-29$^{th}$ July) mostly by Mediterranean recirculations, and Period 5 (29$^{th}$ July-7$^{th}$ August) by Mediterranean recirculations and continental European air masses. The sampling site was therefore influenced by long-range North Atlantic marine emissions and European emissions which were subsequently recirculated over the Mediterranean Sea, with relatively infrequent input from the Sahara. Combined with a sampling period of 8 weeks, it is clear that the site was well placed to provide an insight into the composition of Mediterranean aerosol from diverse sources under a wide variety of meteorological conditions.

### 3.2 ATOFMS particle classes

Twenty-seven distinct ATOFMS particle classes were identified and subsequently grouped into 8 general "categories" for clarity based on the dominant marker ions; EC-rich (53% of total mass spectra), K-rich (32%), Na-rich (8%), Amines (4%), OC-rich (2%), V-rich (1%), Fe-rich (0.2%) and Ca-rich (0.2%). The contribution to total particle number and mass concentration of all particle classes can be found in Table 2.



While the aerosol mixing state is varied, only a few particle classes represent the bulk of un-scaled particle numbers. The dominant classes are *EC-SO$_X$*, *K-SO$_X$* and *K-EC-Oxalate*. The Na-rich category is dominated by *Sea salt-aged* particles, the EC-rich category by *EC-SO$_X$* particles, the K-rich category by *K-SO$_X$* particles, the Amines category by *K-TMA* and *EC-TMA* particles, the OC-containing category by *OC* particles and the V-rich category by *V* particles.

### 3.3 Mass Concentrations

Reconstructed ATOFMS mass concentrations were compared with those for PM$_{10}$, total PILS-IC species (PM$_{10}$) and PM$_1$ (total ACSM species + BC), shown in Figure 2. Reconstructed ATOFMS particulate mass accounted for 70-90% of PM$_{10}$ mass for most of the sampling period. Note that ATOFMS particles account for relatively little of the PM$_{10}$ mass during periods when sea salt and dust are abundant, which is expected given the upper size limit (3 μm) of the instrument and the low detection efficiency for supermicron particles (Cahill et al., 2014). Total ATOFMS reconstructed mass concentrations were found to correlate well with total ACSM mass concentrations (R$^2$ =0.71), and moderately with mass concentrations of PILS-IC SO$_4^{2-}$ and NH$_4^+$ (R$^2$ =0.58, 0.44), the ACSM factors LV-OOA (R$^2$ =0.59) and SV-OOA (R$^2$ =0.46), BC (R$^2$ =0.55) and PM$_1$ (R$^2$ =0.44).

ATOFMS reconstructed mass concentrations were dominated by EC-rich particles (52%), followed by K-rich (25%), Na-rich (12%), Amines (3%), OC-rich (3%), V-rich (3%), Ca-rich (1%) and Fe-rich (1%). The dominance of EC and K-rich ATOFMS particles does not suggest that PM$_{2.5}$ mass is comprised mostly of EC and K. These species are simply used as markers (and for naming conventions) in ATOFMS analysis for fossil fuel and biomass combustion, while truly quantitative measurements by the PILS-IC and ACSM indicate that most of the PM$_{10}$ mass was comprised of organics (36%), sulfate (16%) and ammonium (10%). A detailed discussion follows in Section 3.4.1.

### 3.4 Particle Sources

Four general sources of ATOFMS PM$_{2.5}$ particles were identified during ADRIMED and SAF-MED, namely regionally transported combustion, local biomass burning, marine, and shipping. Composition of the particle classes that may have originated from these sources, and comparison of their mass concentrations with other measurements is discussed in the following sections.

#### 3.4.1 Regionally Transported Combustion

Twelve ATOFMS particles classes were identified as originating from regionally transported combustion, all found in the following 3 categories; EC-rich, K-rich and Amines. Average mass spectra for these particles classes are shown in Figure 3. All of the EC-rich classes were characterised by elemental carbon fragments ions $^{12,24,36\cdots}$C$^+$ in the positive mass spectra. *EC-SO$_X$* and *EC-Oxalate* did not contain detectable $^{39}$K$^+$, indicating they most likely arise from fossil fuel combustion (oil burning or traffic). *EC-K*, *EC-K*-SO$_X$ and *EC-K*-Oxalate were characterised by a stronger signal



for $^{36}C_3^+$ relative to $^{39}K^+$; similar particles have previously been attributed to domestic coal combustion (Healy et al., 2010), although other sources are certainly possible. *K-EC-NO$_X$*, *K-EC-SO$_X$* and *K-EC-Oxalate* produced stronger signals for $^{39}K^+$ relative to $^{36}C_3^+$, a pattern usually associated with biomass burning particles. Sulfate ($^{97}HSO_4H^-$) dominated the SO$_X$ classes, but was also present to a lesser extent in the other six EC-rich classes. Despite the average spectra for the EC-rich

classes showing large signals for sulfate and nitrate ($^{46}NO_2^-$, $^{62}NO_3^-$) most of the particles in these classes produced no negative ion spectra or only weak negative ion signals. It is therefore not possible to definitively describe their anion mixing state; however certain conclusions can still be drawn from their temporal profiles. Less nitrate, or weaker signals for these species, relative to sulfate was expected given the high ambient temperatures; nitrate is usually mixed with EC in the form of am-

monium nitrate, which is more volatile than ammonium sulfate (Querol et al., 2009; Sciare et al., 2008). A smaller signal for MSA, $^{95}(CH)_3SO_3^-$ (Neubauer et al., 1997), relative to sulfate was found in the *K-EC-SO$_X$* class, indicating processing with marine emissions (Gaston et al., 2010). The oxalate classes are characterised by their signal at m/z -89, a marker for deprotonated oxalic acid (Yang et al., 2009) and aged aerosol. Very small signals for ammonium ($^{18}NH_4^+$), not marked in the mass

spectra, were found in all classes with the exception of the *EC-Oxalate* class.

The positive ion mass spectra for *K-CN* and *K-SO$_X$* particles are exclusively dominated by $^{39}K^+$, typical of biomass burning particles detected by ATOFMS (a.R. Lea-Langton et al., 2015; Pratt et al., 2010; Qin and Prather, 2006; Silva et al., 1999). Sulfate ($^{97}HSO_4H^-$) dominates the negative ion mass spectra of the *K-SO$_X$* particles, with additional signals for MSA ($^{95}(CH)_3SO_3^-$) and ox-

alate ($^{89}(COO)_2H^-$). $^{26}CN^-$, internally mixed carbon and nitrogen, probably in the form of nitrogen-containing organic compounds (Silva et al., 1999), dominates the negative ion mass spectra of *K-CN* particles. EC fragments were found in all K-rich negative ion mass spectra, indicating a biomass combustion source.

Trimethylamine (TMA, $^{59}(CH_3)_3N^+$,) was the most abundant alkylamine marker ion in the three

Amine particle classes. Also present in all three classes was a marker ion for protonated dimethylamine (DMA, $^{46}(CH_3)_2NH_2^+$). A comparison of ATOFMS datasets obtained in Cork, Paris, Zurich, Dunkirk and Corsica (ADRIMED and SAFMED) found this ion only in the latter two datasets (Healy et al., 2015). Ammonium ($^{18}NH_4^+$) was also found in all three Amine classes. The *K-TMA* class was dominated by $^{39}K^+$, indicative of biomass burning, while *EC-TMA* particles produced $^{12,36}C_3^+$ sig-

nals, indicating fossil fuel combustion origins. *OC-TMA* particles are characterised by strong $^{39,41}K^+$, OC ($^{27}C_2H_3^+$) and oxidised OC ($^{43}C_2H_3O^+$) signals, suggesting biomass burning sources and atmospheric processing during transport to the site. Sulfate ($^{97}HSO_4H^-$) and nitrate ($^{46}NO_2^-$, $^{62}NO_3^-$, $^{125}(NO_3)_2^-$) were found in the average negative mass spectra of *EC-TMA* and *OC-TMA*, however only 0.3% and 18% of these particles actually produced negative mass spectra and *K-TMA* parti-

cles produced none. Alkylaminium sulfate particles have been shown to readily absorb water at low relative humidities (<45%) (Chan and Chan, 2013; Hu et al., 2014), and particle-bound water has



been shown to suppress negative ion formation in mass spectrometers (Neubauer et al., 1997, 1998). The proportion of Amine particles with negative mass spectra is low, in contrast to other particle classes which produced signals in up to 100% of their negative mass spectra (*K-EC-SO$_X$*, *K-SO$_X$-*

*Oxalate*). This suggests particle-bound water could have had a significant effect on negative ion formation for these particles. The average negative mass spectra should therefore not be considered as representative of every particle in these classes. Healy et al. (2015) previously assigned m/z -95 in amine-containing particles from this dataset to MSA. However, closer inspection of individual spectra suggests that at least some of the signal at -95 arises from miscalibrated negative ion mass spectra,

and is in fact associated with sulfate. Although m/z -95 cannot therefore be definitively assigned to MSA here, internal mixing of MSA and TMA is possible, as indicated by laboratory studies (Chen et al., 2015a). Furthermore, MSA was also measured in bulk PM$_{10}$ during this campaign using the PILS-IC instrument, and is thus inevitably contained in a fraction of particles at this location. Temporal profiles for hourly summed particle numbers for the regionally transported combustion classes

are shown in Figure 4. There are three general temporal profiles; those of *EC-SO$_X$*, *EC-K-SO$_X$*, *EC-K* and *EC-Oxalate*, those of *K-SO$_X$*, *K-EC-SO$_X$*, *K-CN* and *K-EC-Oxalate*, and finally those containing TMA. The EC-rich particle classes dominated Period 1 and were considered markers for fossil fuel combustion because most of these did not contain detectable $^{39}$K$^+$. Their particle numbers remained relatively similar throughout the campaigns. For the first half of Period 1 regional air masses passed

over industrialised parts of France (Marseille) and Northern Italy (Po Valley) before reaching the site, while during the latter half of this period these air masses were then recirculated over the western Mediterranean (SI Figure 1 shows the air mass back trajectories calculated for this period using the HYSPLIT model).

Particle numbers for all EC-rich and major K-rich classes decreased noticeably during Periods 2

and 5, which were influenced by synoptic scale air masses from the North Atlantic, effectively removing aerosol accumulated during the previous periods. On the other hand, K-rich particles, markers for biomass combustion, dominated Periods 3 and 4. Fires were detected by MODIS (Figure 5) over northern Italy, Ukraine and Russia throughout the sampling period and the site was heavily influenced by air masses passing over this region (Figure 1), which could explain the constant presence

of K-rich particles in the background aerosol. The increase in these particles during Period 3 may be explained by longer residence times of air masses over southern France and northern Italy relative to Period 1, and particle numbers then began to decrease when trajectories from the North Atlantic and UK arrived at the end of Period 3.

A significant increase in fires was detected around the Black Sea from 10[th] July until the end of

the sampling period (Figure 5). The burning of wheat residuals has been previously documented in the eastern Mediterranean and contributed at least 30% of EC and OC measured during similar time periods between 2001-2006 (Sciare et al., 2008). For ~4 days during Period 4, air masses from over the Black Sea and eastern Europe influenced the site, followed by extensive stagnation and





recirculation of that air over the western Mediterranean. These observations coincided with a further

increase in K-rich particle numbers relative to Period 3.

As shown in Figure 4, the three Amine particles classes, *K-TMA*, *EC-TMA* and *OC-TMA*, presented similar temporal profiles in Period 4 to those of major EC-rich and K-rich particles classes. Numbers of *K-TMA* and *EC-TMA* particles peaked during Period 4, while *K-TMA* particles dominated over *EC-TMA* particles during Period 3, as was the case for the major K-rich and EC-rich

classes. The dominant Amine class, *K-TMA*, correlated well with two major classes, *K-SO$_X$* and *K-EC-Oxalate*, particularly during Period 4 ($R^2$=0.76, 0.70 respectively). Partitioning of alkylamines from the gas phase to particles has been found to be enhanced during periods of high relative humidity or fog events, with uptake increasing with aerosol acidity Rehbein et al. (2011). No association between any of the Amine classes and local relative humidity was found in this case, suggesting that

this effect is not relevant close to the site but may have played a role close to the point of emission or during transport of the Amine particles to the site. It also provides further evidence that the Amine particles were not formed in the local environment.

To investigate how the dominant ATOFMS particle categories (EC and K-rich) compared with the dominant bulk PM$_{10}$ (excluding sea salt) species, hourly mass concentrations of PM$_{10}$, BC, ACSM

species (SO$_4^{2-}$, NH$_4^+$ and SVOOA) and reconstructed mass concentrations for ATOFMS EC-rich and K-rich particle categories (combined particle classes) were plotted (Figure 6). EC and K-rich mass concentrations have been stacked to compare their values with the contributions from BC, SV-OOA, LV-OOA, SO$_4^{2-}$ and NH$_4^+$ (also stacked). For the full sampling period reconstructed mass concentrations of all ATOFMS EC-rich particle classes correlated well with ACSM SO$_4^{2-}$ ($R^2$=0.61),

NH$_4^+$ ($R^2$=0.62) and the LV-OOA factor ($R^2$=0.59) mass concentrations (which accounted for the largest proportion of organics mass, 54%), OPS (for the channel 0.3-0.579 μm) number concentrations ($R^2$=0.69) and moderately with PM$_1$ ($R^2$=0.46) and BC ($R^2$=0.50) mass concentrations. Reconstructed mass concentrations for all K-rich classes (including local combustion) correlated with the same species but only moderately ($R^2$ ranged from 0.3-0.5). Individual particle classes did

not produce stronger correlations, suggesting that no single class was an important contributor of PM$_{2.5}$ composition.

The average mass spectra of ATOFMS EC and K-rich classes showed sulfate, nitrate, oxalate and MSA were present. From the PILS-IC and ACSM measurements it was clear that the dominant secondary inorganic ions were sulfate (19% of PM$_1$, 16% of PM$_{10}$) and ammonium (10% of PM$_1$,

10% of PM$_{10}$), with nitrate contributing relatively little to both PM$_1$ (4%) and PM$_{10}$ (4%) mass. K$^+$, oxalate and MSA contributed less again to PM$_{10}$ (0.4, 0.1 and 0.2% respectively). K$^+$ is readily ionised by the ATOFMS desorption/ionisation laser so its prevalence in ATOFMS particles is not necessarily representative of its mass concentration (Gross et al., 2000a).

The high level of agreement between mass concentrations of EC-rich+K-rich particles and BC+SV-

OOA+LVOOA+SO$_4^{2-}$+NH$_4^+$, shown in a scatter plot in Figure 6, suggests those particles were com-





prised to a considerable degree of LV-OOA, consistent with the ageing of the aerosol particles during regional transport, and SV-OOA, which suggests interaction with locally formed organic aerosol; details of organic aerosol formation and sources during the campaigns can be found in Michoud et al. (2016). However, no significant OC signals were found in any of the EC-rich or K-rich classes. Par-
ticles which did produce strong signals for OC ions contributed relatively little to total ATOFMS particle numbers. Similarly to $^{39}$K$^+$, the ATOFMS favours ionisation of EC over OC (Ferge et al., 2006; Silva and Prather, 2000), which results in weak signals for OC in particles which also contain EC or K$^+$. This may account for the under-representation of organic aerosol in the ATOFMS measurements.

### 3.4.2 Local biomass burning

The sampling site on Corsica was chosen for its negligible local anthropogenic sources relative to the regional background. However, few local sources did influence the site, though these did not contribute significantly to particle number or mass concentrations. Local combustion events were detected in the form of *K-NO$_X$*, K-OCNOX, and *K-OC-SO$_X$* particles, and distinguished themselves
from the dominating regional aerosol by occurring mostly during Periods 1 and 2 over 5-7 hour events (Figure 7). The first event was observed on the 12$^{th}$ June from 13:00-18:00 UTC and its source as biomass burning (in the form of vegetation trimmings) was visually confirmed on the slopes northeast of the site.

The composition of these particles is consistent with this observation; K$^+$ is a common marker
for biomass combustion, which typically also produces organic aerosol (Pagels et al., 2013; Silva et al., 1999). Average mass spectra for local biomass burning particles are shown in Figure 8. Nitrate ($^{46}$NO$_2$$^-$, $^{62}$NO$_3$$^-$) dominated the negative mass spectra of *K-NO$_X$* particles, but they also contained sulfate ($^{97}$HSO$_4$H$^-$), nitrogen-containing organic compounds ($^{26}$CN$^-$), EC ($^{24,36,48}$C$^-$) and oxygen ($^{16}$O$^-$). K-OC-NO$_X$ and K-OC-SO$_X$ are characterised by a large signal for $^{39}$K$^+$ (confirmed by
the prominent signal for the $^{41}$K$^+$ isotope, hydrocarbon fragments ($^{27}$C$_2$H$_3$$^+$, $^{29}$C$_2$H$_5$$^+$, $^{51}$C$_4$H$_3$$^+$, $^{63}$C$_5$H$_3$$^+$) in the positive mode and strong signals for $^{43}$C$_2$H$_3$O$^+$, a marker for oxidised organic aerosol (Silva and Prather, 2000). Sulfate was found in both OC-rich classes, but dominated the negative mass spectra of *K-OC-SO$_X$* particles. K-OC-SO$_X$ particles also exhibited a small MSA ($^{95}$(CH)$_3$SO$_3$$^-$) signal, indicating at least some mixing with marine biogenic emissions prior to detec-
tion. Nitrate dominated the *K-OC-NO$_X$* class and was also present to a lesser extent in the *K-OC-SO$_X$* classes.

Garden waste biomass was frequently burned in the surrounding villages during June; such combustion was prohibited from July onwards which explains the lack of similar local events. No local wildfires or controlled agricultural burning was noted during the sampling period. Between 27$^{th}$ June
and 1$^{st}$ July, *K-NO$_X$*, *K-EC-NO$_X$* and possibly *K-OC-SO$_X$* were of regional biomass burning origin, because their temporality was noticeably different to the preceding short local events; this period





was influenced by short-range air masses residing over southern France and northern Italy. Peak aerodynamic diameters for these particles were also larger during this period compared to previous local combustion events; 700-900 nm ($D_a$) versus 300-500 nm. This represents a significant amount

of growth through ageing, assuming these particles were of a similar size at their point of origin to those burned locally in Corsica. *K-NO$_X$* and *K-OC-SO$_X$* particles were also present throughout both campaigns, in low numbers outside of events, indicating persistent regional sources.

### 3.4.3 Sea Salt

ATOFMS sea salt particles were separated into two classes; fresh and aged. Their composition and
comparison with other sea salt measurements (e.g. Na and Cl mass concentrations from PILS-IC) are the subject of a detailed study of primary marine aerosols during ADRIMED (Claeys et al. 2016), so will only be discussed briefly here. Both sea salt classes are typical of those observed in other coastal/marine environments (Gard et al., 1998; Dall'Osto et al., 2004; Healy et al., 2010). Average mass spectra are shown in Figure 8. The positive modes for both fresh and aged particles are
similar and are characterised by sodium ions ($^{23}Na^+$, $^{46}Na_2^+$, $^{62}Na_2O^+$, $^{63}Na_2OH^+$ and $^{81,83}Na_2Cl^+$) and $^{39}K^+$. Fresh and aged sea salt particles were differentiated by their negative mass spectra, which showed peaks for $^{16}O^-$, $^{35,37}Cl^-$, nitrate ($^{46}NO_2^-$, $^{62}NO_3^-$) and $^{93,95}NaCl_2^-$ for fresh sea salt particles, while the signals for nitrate dominate the aged sea salt negative mode and sodium chloride adducts are virtually absent. The absence of NaCl ions and strong nitrate signals indicates extensive replace-
ment of $Cl^-$ by $NO_3^-$ (Gard et al., 1998), while the presence of nitrate in the negative mass spectra of the fresh sea salt particles suggests that these are not truly fresh but have also undergone some Cl replacement.

Temporal profiles for ATOFMS sea salt particle numbers, OPS number concentrations, PILS-IC sea salt aerosol (SSA, calculated using SSA=[Cl$^-$]+[Na+]×1.47; Bates et al., 2012) and ATOFMS
fresh sea salt mass concentrations are shown in Figure 9. The two ATOFMS sea salt classes presented noticeably different temporal profiles. Aged sea salt was present consistently in the background throughout both campaigns, while fresh sea salt was detected mostly during short periods (20-26th June, 30th July). This coincided with increases in OPS number concentrations in the 0.579-2.156 μm range and PILS-IC SSA. Correlation between ATOFMS fresh sea salt mass and SSA is
particularly strong ($R^2$=0.81) for the sea salt event during Period 2. Reconstructed mass concentrations for fresh sea salt particles accounted for 50-80% of SSA during the main event. This and a strong correlation between ATOFMS fresh sea salt mass concentrations and 0.579-2.156 μm particles ($R^2$=0.81) suggests a significant amount of fresh sea salt was in the PM$_{2.5}$ fraction; an estimated 30% of PM$_{10}$ SSA mass was accounted for by PM$_{2.5}$ ATOFMS fresh sea salt during the 5-day event
in Period 2.




### 3.4.4   Mineral Dust

Three prominent mineral dust events were characterised by increases in $Ca_2^+$ mass concentrations
and 2.156-8.032 μm particle number concentrations during 12-13[th] June, 17-19[th] June and 23-26[th]
June (Figure 9). The third event coincided with the main sea salt event and was also distinguished
by contributions of $K^+$. The 17-19[th] June event is likely related to a moderate African dust event that
passed over Corsica, as shown by MSG\SEVIRI aerosol optical depth (SI Figure 2). The two other
periods appear due to local dust. Two particle classes were identified as potential mineral dust by
the ATOFMS; Fe and Ca. Average mass spectra are shown in Figure 8. Fe and Ca, along with Al
and aluminosilicates are typical dust tracers which produce ions detectable by ATOFMS (Guazzotti
et al., 2001; Silva et al., 2000; Sullivan et al., 2007).

K-rich dust is also a possibility. The Fe particles detected in Corsica were internally mixed with
$^{39}K^+$, $^{23}Na^+$, $^{27}Al^+$, sulfate ($^{97}HSO_4H^-$) and nitrate, although only a weak signal for $^{27}Al^+$ and no
aluminosilicate signals (e.g. $^{43}AlO^-$, $^{59}AlO_2^-$, $^{60}SiO_2^-$, $^{76}SiO_3^-$, $^{77}HSiO_3^-$, $^{103}AlSiO_3^-$) were found,
which could also suggest industrial origins (Corbin et al., 2012; Dall'Osto et al., 2008; Zhang et al.,
2009). Ca particles are dominated by $^{40}Ca^+$, with weaker signals for $^{56}CaO^+$ and $^{96}CaO_2^+$. 34% of
all Ca particles produced positive EC ions, which suggests a vehicular traffic source (Gross et al.,
2000b; Silva and Prather, 1997; Song et al., 2001) to and from the site and from a few local villages
whose tourist population increases during the summer.

The Fe and Ca classes only contributed a small number of particles (0.2% each) relative to the
total number ionised by the ATOFMS and no agreement between these potential dust particles and
PILS-IC or OPS measurements was found. This indicates that mineral dust was not characterised
well by the ATOFMS. Indeed, the dust particle mass-size distribution is mainly in the >$PM_{2.5}$ frac-
tion, but submicron particles dominate the dust particle number size distribution (see  Guieu et al.,
2010)(Gomes et al., 1990) as confirmed by in situ and column-integrated particle size distribution
measurements during the campaign (Denjean et al., 2016; Renard et al., 2016).

### 3.4.5   Shipping

Two V-rich particle classes (*V* and *EC-V*, 2% and 1% of ATOFMS mass respectively) were identified
as originating from heavy fuel oil combustion; both contained $^{51}V^+$, $^{67}VO^+$, $^{56}Fe^+$, $^{58}Ni^+$ and sulfate
($^{97}HSO_4H^-$), which are typical markers for particles emitted by ships or oil refineries (Ault et al.,
2009; Healy et al., 2009). A small signal for MSA ($^{95}(CH)_3SO_3^-$) was present in *EC-V* particles.
Internally mixed sodium, potassium, calcium, vanadium, nickel and iron particles have also been
observed in ship exhaust particles using off-line TEM-EDX and two-step laser mass spectrometry
(L2MS) (Moldanová et al., 2009). Small signals for $^{39}K^+$ and $^{23}Na^+$ were found in the Corsica V-
rich particles, but none for $^{40}Ca^+$. Heavy fuel oil combustion aerosols have a strong presence in the
Mediterranean (Becagli et al., 2016; Pey et al., 2010; Querol et al., 2009). There are more than 15





passenger ferry lanes incurring shipping traffic around the northern tip of the island; the closest pass is 16.5 km north and 12.5 km east of the site (Figure 10). Ferries travelling around the northern tip of Corsica Island, take approx. one hour to reach Bastia and between all five ferry companies ~50 sailings take place per week. Both V-rich classes displayed strong north-westerly and southwesterly

wind dependences (Figure 10), consistent with the distribution of most ferry lanes (Figure 10) V-rich particles were identified as aged regional emissions. Most freshly emitted shipping particles are typically less than 300 nm $D_a$ (Healy et al., 2009). However, all shipping particles detected during this campaign had diameters larger than 300 nm $D_a$. Furthermore, it was unlikely that any fresh heavy oil combustion emissions were observed with the ATOFMS as the closest ferry lane is at least

12.5 km from the site. There are also no local power plants or refineries that could generate similar particles. *EC-V* and *V* particle numbers consistently featured a mode around 740 nm $D_a$ indicating that the observed particles were aged to some degree.

### 3.5  Aged Particle Markers

As a substantial number of ionised ATOFMS particles produced low intensity signals in their neg-

ative ion mass spectra, or none at all, the average mass spectra are most representative of those species that ionised most efficiently i.e. nitrate and sulfate. Ions that usually produce relatively small signals, such as oxalate or MSA, were expected to be under-represented in the average mass spectra so an additional querying approach was taken to examine the mixing state of these species; particle numbers for classes found to contain these species are shown in Table 2. Oxalate ($^{89}(COO)_2H^-$) and

MSA ($^{95}(CH)_3SO_3^-$) ions were queried for peak height between 1 and 5000, to include all mass spectra containing these species. These particles were then clustered using the K-means algorithm to produce particle classes.

### 3.5.1  Oxalate

The mixing states of particles containing oxalate as identified by single particle techniques is var-

ied, and indicates oxalate formation either in fog/cloud processing or photochemical oxidation of biogenic and anthropogenic VOCs. Oxalate has been found in biomass burning particles (Healy et al., 2010; Yang et al., 2009), mixed with industrial combustion particles containing Pb and Zn (Moffet et al., 2008)(Moffet et al., 2008a, 2008b), in aged sea salt (Yang et al., 2009) and in aged carbonaceous particles containing highly oxidised organics, non-oxygenated organics and amines

(Pratt et al., 2009; Qin et al., 2012). Oxalic acid has been found preferentially enriched on Asian mineral dust over carbonaceous particles (Sullivan and Prather, 2007), while Fitzgerald et al. (2015) characterised cloud processed African dust as rich in sulfate and oxalate.

In our study, oxalate was found in ~9600 particles, i.e. 0.8% of the total particles ionised. The mixing states derived from this query are similar to those produced from general clustering and



are varied, suggesting that poor ionisation efficiency did not prevent oxalate from being detected in certain types of particles.

From the querying approach it was apparent that K-rich particles produced more signals for oxalate than the more abundant carbonaceous particles. This could indicate preferential partitioning to K-rich particles or more extensive oxidation of the OC in those particles versus EC-rich ones.

52% of the queried oxalate particles were dominated by $^{39}K^+$ and sulfate (~90% of all the queried particles contained sulfate), supporting the identification of the *K-SO_X-Oxalate* class from the general approach. Particles most similar to *K-NO_X* accounted for a further 33% and produced signals for nitrate, EC, CN, CNO and sulfate. In contrast only 2% of queried oxalate particles produced EC-rich positive mass spectra. Particles with typical dust tracers (Fe, Ca, Al, aluminosilicates) ac-

counted for 7% of the queried particles; this fraction applies to $PM_{2.5}$ but would certainly be larger for $PM_{10}$. The remaining queried oxalate particles were classified as *aged sea salt* (3%), *OC* (2%), *V* (0.7%) and *Cu-Pb* (0.4%). If the queried oxalate particle numbers were considered representative of the whole ATOFMS dataset, then biomass burning emissions play a large role in the fate of particle phase oxalate in the western Mediterranean.

The ATOFMS querying approach indicated a prevalence of oxalate mixed with K-rich particles, however correlations between PILS-IC oxalate and K-rich mass were poor; better correlations were found with EC-rich and K-rich mass concentrations combined ($R^2$=0.55), suggesting either that more EC particles actually contained oxalate than was detected or another particle type transported in the same air mass but which was not detected by the ATOFMS. Moderate correlations were also found

between oxalate and LV-OOA ($R^2$=0.54), WSOC ($R^2$=0.55) and BC ($R^2$=0.55) mass concentrations (for the period 21st June-4th August); hourly mass concentrations for these species are shown in Figure 11 (mean concentration of oxalate: 9.8 ng/m$^3$). Oxalic acid is often the single most abundant water-soluble organic compound identified in ambient aerosols (Yu et al., 2005), which explains the agreement between oxalate and WSOC mass concentrations. While the humidity was relatively high

(average of 70%) throughout the two campaigns, so too was the solar radiation, with few instances of cloud or fog formation at the site. The association with SV-OOA also supports photochemical oxidation as the dominant oxalate formation mechanism.

### 3.5.2 MSA

MSA is a well-established tracer for marine phytoplankton activity (Hallquist et al., 2009; Gaston

et al., 2010), formed from heterogeneous OH (daytime) and homogeneous $NO_3$ (nighttime) oxidation of DMS, the enzymatic cleavage product of dimethylsulfoniopropionate (DMSP), a compound produced by oceanic phytoplankton. It is therefore a good indicator of biogenic (marine) sulfate and its presence in aerosols typically indicates that they have undergone some marine transport, rather than being produced locally (Gaston et al., 2010). While MSA has been proposed to primarily con-

tribute to particle growth in the atmosphere (Kreidenweis et al., 1989; Wyslouzil et al., 1991a, b),



there is evidence that MSA can also contribute to new particle formation (Dawson et al., 2012; Willis et al., 2016). Recently, Sellegri et al. (2016) showed that iodine-containing species are likely precursors to new particle formation in the Mediterranean; a query of the ATOFMS ADRIMED and SAF-MED dataset for iodine (m/z -127) only returned 34 particles containing this ion. Iodine is

rarely reported in ATOFMS studies and even then it is usually in low numbers of particles; Beddows et al. (2004) found ~3600 $PM_{2.5}$ ATOFMS particles containing iodine at a rural background site in the UK. Both MSA and sulfate influence particle hygroscopicity, meaning that the enhanced production of either of these species by anthropogenic particle types could have implications for subsequent cloud droplet formation in both marine and inland environments (Lee et al., 2010; O'Dowd et al.,

2004).

MSA was identified in 2700 particles (0.2% of total particles ionised). Sulfate was also found in all of these. The mixing states derived from the querying approach largely agrees with those from the general approach. Similar to oxalate, enhanced partitioning of MSA was observed for biomass combustion particles. Particles similar to the *K-SO$_X$*, *EC-K-SO$_X$* and *K-EC-SO$_X$* classes accounted

for 45%, 25% and 11% of the MSA queried particles. *EC-V, V* and *Cu-Pb* classes contributed 11%, 4% and 1% respectively.

The preference for partitioning to combustion particles is in contrast to findings from Riverside, California (Gaston et al., 2010) where only small fractions of carbonaceous ATOFMS particle types contained MSA, and Cork, Ireland (Healy et al., 2010), where only one ATOFMS particle class

of several carbonaceous classes identified was internally mixed with MSA. It has been suggested that accumulation of secondary species during transport over urban regions can potentially mask the detection of MSA by ATOFMS in carbonaceous particles (Pratt and Prather, 2009).

In addition, it has been demonstrated that the oxidation of biogenically emitted DMS to form MSA, can be catalysed by vanadium, which has also been shown to enhance the conversion of

anthropogenically produced $SO_2$ to sulfate (Ault et al., 2010). About 40% of OC-V-sulfate (residual fuel combustion primarily from ships) particles and 33% of aged sea salt particles in Riverside contained MSA. No MSA could be clearly identified in aged sea salt particles in Corsica; m/z -95 was only found in fresh sea salt particles and was likely a result of $NaCl_2$ as it was present in the typical isotopic ratio i.e. its signal was smaller than m/z -93. This is again in contrast to the study

in Cork harbour (Healy et al., 2010), where none of the V-rich shipping classes were associated with MSA. However, the authors note that the sampling site in Cork Harbour was very close to shipping berths, so most shipping particles detected were expected to be freshly emitted. *K-OC-SO$_X$* particles, identified with the general clustering approach, produced average mass spectra with MSA signals. This was not supported by the querying approach; no OC-rich particles were found to

contain MSA, or at least so few as to not resolve into their own clusters. No Amine particles were identified in the MSA query. As discussed earlier, the presence of MSA in TMA and other amine-containing particles is possible, as indicated by laboratory studies of particle formation and growth





from reactions between MSA, TMA (or methylamine and DMA) and water (Chen et al., 2015a, b). Amines have been frequently observed in marine aerosol (Facchini et al., 2008; Gaston et al., 2010; Müller et al., 2009; Sorooshian et al., 2009), and particulate amine levels have been found to correlate with particulate MSA levels (Sorooshian et al., 2009). Facchini et al. (2008) observed that MSA, DMA, and diethylamine were the most abundant organic species detected in fine particles in the North Atlantic during periods of high biological activity.

The remaining 3% of MSA queried particles was accounted for by particles containing dust tracers (Fe, Ca, Al). The Riverside study found that no MSA was found on submicron dust and only 3% of supermicron dust contained MSA (Gaston et al., 2010). The authors expected this since the dust they observed was locally produced, unlike MSA. In contrast, internally mixed OC, sea salt, sulfate, titanium (dust) and MSA formed particles 1 – 2 μm in size at Mace Head during the EUCAARI project Dall'Osto et al. (2010) and were associated with a period of subtropical maritime air originating from the Azores high-pressure region. The number of dust particles identified with the MSA query in this work were statistically too few to consider submicron/supermicron ratios; from these previous studies it is not unusual to detect MSA on dust particles, however it is unlikely that they represent significant surfaces for MSA to condense onto as most of the MSA mass measured in the Mediterranean to date has been found in the submicron fraction. This observation is also echoed by Gaston et al. (2010); 67% of the submicron particles in Riverside contained MSA.

In Riverside MSA-containing particles were also associated with fog processing markers $^{81}HSO_3^-$ and hydroxymethanesulfonate (HMS, $^{111}HOCH_2SO_3^-$), highlighting aqueous phase chemistry as an important pathway in MSA formation (Bardouki and Rosa, 2002), as well as the hygroscopic nature of MSA (Barnes et al., 2006). Other studies correlated HMS with relative humidity (RH) during stagnant fog events (Whiteaker and Prather, 2003), however in Riverside the HMS correlation with MSA suggested HMS formation was not due to local increases in RH; rather MSA-particles had undergone aqueous phase processing either in the marine environment or during subsequent transport. Unfortunately queries for m/z -81 and -111 did not return a statistically useful number of particles, and the summed signal for both of these groups was too low to produce a statistically useful time series.

Hourly PILS-IC mass concentrations of MSA (average of 21 ng/m$^3$; lower than those measured in Paris; average of 122 ng/m$^3$; Crippa et al. (2013) compared to Mace Head, Ireland and Erdemli, Turkey) shown in Figure 12, did not correlate with any ATOFMS particle numbers or mass concentrations, or with any other measurements over the whole sampling period. However, some correlations were found for certain periods; from 7-15$^{th}$ July MSA mass agreed well with that of PILS-IC NO$_3^-$, while from 23$^{rd}$ July – 3$^{rd}$ August moderate correlations were found with PILS-IC NO$_3^-$ (R$^2$=0.53), SO$_4^{2-}$ (R$^2$=0.48) and NH$_4^+$ (R$^2$=0.54), and ACSM NO$_3^-$ (R$^2$=0.46). Increases in MSA mass coincided with sea salt events during Period 2 (Figure 12), although there were no good corre-



lations with SSA concentration, suggesting MSA was not present on sea salt particles but formation
was enhanced by the influx of marine air masses.

## 4   Conclusions

As part of ChArMEx, two special observation periods on Corsica Island aimed to understand how the
physical, chemical, optical properties and vertical distribution of aerosols affect the Mediterranean
climate (ADRIMED), as well as develop a better understanding of the origins and particle proper-
ties of secondary organic aerosols (SAF-MED). Chemical composition is critical to achieving these
aims. A single particle mass spectrometer (ATOFMS) provided detailed information on the mixing
states and thereby sources of background aerosol in the western Mediterranean. Airmass trajecto-
ries and concurrent observations at the site were used to interpret ATOFMS observations. Overall,
27 distinct ATOFMS particle classes were identified from 1.2 million single particle mass spectra
and grouped into 8 general categories: EC, K-rich, Na-rich, Amines, OC-rich, V-rich, Fe-rich and
Ca-rich. Mass concentrations were reconstructed for the ATOFMS particle classes and were in good
agreement with other quantitative measurements ($PM_1$, ACSM species, BC). Total ATOFMS esti-
mated mass ($PM_{2.5}$) accounted for 70-90% of $PM_{10}$ mass, most of which was comprised of region-
ally transported aerosols containing fossil fuel combustion (EC-rich) particles, and K-rich particles
from biomass burning in northern Italy and the region surrounding the Black Sea, the accumulation
of which was favoured by repeated and extended periods of air mass stagnation over the western
Mediterranean. Amine-containing particles were also assigned to regionally transported combus-
tion sources, both fossil fuel and biomass burning. Previous studies of aminecontaining particles
found a strong dependence on relative humidity; this was not the case during these two campaigns,
suggesting these particles were not formed locally. Three other sources were also identified by the
ATOFMS: local biomass burning, marine and shipping. Local biomass burning particles contributed
little to $PM_{2.5}$ particle numbers and mass concentrations but were easily distinguished from regional
combustion particles; no local sources of fossil fuel combustion were identified. Although the local
emissions did not contribute significantly to particle number or mass concentrations the observa-
tions serve to highlight the ability of single particle measurements to distinguish between local and
regional aerosol sources. Marine emissions comprised fresh and aged sea salt, the former detected
mostly during one 5-day event and the latter detected throughout the sampling period. Mineral dust
was not efficiently detected by the ATOFMS. Shipping particles, identified using markers for heavy
fuel oil combustion, were identified as aged regional emissions which made only a small contribu-
tion to $PM_{2.5}$ particle numbers and mass concentrations. A query of the mixing states of oxalate, a
photochemically aged aerosol marker, and MSA, a biogenic marine emissions marker showed that
the majority of particles containing oxalate also contained K and sulfate, indicative of aged biomass
burning emissions. MSA was also strongly associated with biomass burning particles and to a lesser




extent with Shipping particles – probably related to transport time in the marine boundary layer.

Quantitative measurements by TEOM demonstrated that $PM_1$ particles accounted for most of $PM_{10}$ mass concentrations over the whole sampling period. ACSM ($PM_1$) and PILS-IC ($PM_{10}$) sulfate and ammonium mass concentrations were very similar, indicating most of the mass of these species was in the $PM_1$ fraction. Accordingly, organics (36%), sulfate (16%) and ammonium (10%) constituted most of the $PM_{10}$ mass. Mass concentrations of EC and K-rich particles were in good agreement

with those of ACSM sulfate, ammonium and the LV-OOA factor (which accounted for 54% of the organics), and BC. ATOFMS mass spectra provided valuable source markers, allowing the identification of fossil fuel and biomass burning combustion sources. Combined, this information shows these sources provided the primary particles, containing EC and OC, which then accumulated ammonium, sulfate and alkylamines during regional transport. The Mediterranean is a crossroad for air

masses transporting different types of aerosols from natural and anthropogenic origins. Identifying these sources and apportioning aerosol mass to them is a key component of future work to mitigate their effects on the Mediterranean climate.

*Author contributions*. PILS-IC, ACSM, TEOM $PM_{10}$, TEOM $PM_1$ and MAAP data were provided by J. Sciare; SMPS, OPS and meteorological data were provided by Météo-France. J. Arndt prepared

the manuscript with contributions from all co-authors.

*Acknowledgements*. This research has received funding from the French National Research Agency (ANR) projects ADRIMED (grant ANR-11-BS56-0006) and SAF-MED (grant ANR-12-BS06-0013). This work is part of the ChArMEx project supported by ADEME, CEA, CNRS-INSU and Météo-France through the multidisciplinary programme MISTRALS (Mediterranean Integrated Studies aT Regional And Local Scales). The

station at Ersa was partly supported by the CORSiCA project funded by the Collectivité Territoriale de Corse through the Fonds Européen de Développement Régional of the European Operational Program 2007-2013 and the Contrat de Plan Etat-Région. J.Arnd received a fellowship from the Irish Research Council. We gratefully acknowledge the contributions of Dr Ian O'Connor and Dr Eoin McGillicuddy (UCC) who provided excellent support for logistical arrangements and set-up on site.





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


**Table 1.** List of instruments deployed at Cap Corsica during the ADRIMED and SAF-MED field campaigns. See text for instrument acronym details.

| Parameter | Instrument | Make & Model | Temporal resolution | Institution |
|---|---|---|---|---|
| Particle number size distribution (10-500 nm, mobility diameter and 300 nm – 20 μm, optical diameter) | SMPS | TSI Instruments Ltd., DMA model 3080 and CPC model 3010 | 5 min (continuous) | CNRM GAME |
| | OPS | TSI Instruments Ltd., model 3300 | 5 min (continuous) | CNRM GAME |
| $PM_{10}$ and $PM_1$ mass concentration | TEOM, TEOM-FDS | | continuous | LSCE |
| Chemical composition and size distribution of non-refractory and refractory particles (100-3000 nm, vacuum aerodynamic diameter) | ATOFMS | TSI Instruments Ltd., model 3800-100 | continuous | UCC |
| Chemical composition of non-refractory particles (30-1000 nm, vacuum aerodynamic diameter) | ACSM | Aerodyne Research Inc. | 25 min (continuous) | LSCE |
| Black carbon (BC), $PM_{2.5}$ | MAAP | Thermo-Scientific, model 5012 | 5 min (continuous) | LSCE |
| $PM_{10}$ chemical composition ($Na^+$, $Mg^+$, $Cl^-$, $Ca^{2+}$, $K^+$, $NH_4^+$, $NO_2^-$, $SO_4^{2-}$, MSA, Oxalate) | PILS-IC | | 12/18 min (continuous) | LSCE |
| Pressure, temperature, relative humidity, solar radiation, rain, wind speed and direction | Weather station | Campbell Scientific, Model CR1000 | 5 min (continuous) | LSCE |

**Table 2.** Detailed composition of ATOFMS dataset during ADRIMED and SAF-MED, by particle classes.

| Category | Particle Class | No. of Particles | % of Total Ionised | % Particles with negative spectra | No. of Particles containing Oxalate | No. of Particles containing MSA | Assumed Density (g/cm³) | % of Total ATOFMS Mass | Unscaled Peak Aerodynamic Diameter (μm) |
|---|---|---|---|---|---|---|---|---|---|
| EC | EC-SO$_x$ | 329555 | 28 | 3 | | | 1.4 | 22 | 0.74 |
| | EC-Oxalate | 15462 | 1 | 0.5 | | | 1.4 | 1 | 0.72 |
| | EC-K | 40666 | 3 | 16 | 222 | | 1.4 | 4 | 0.76 |
| | EC-K-SO$_x$ | 13627 | 1 | 76 | | 445 | 1.4 | 5 | 0.74 |
| | EC-K-Oxalate | 23399 | 2 | 0.3 | | | 1.4 | 1 | 0.74 |
| | K-EC-NO$_x$ | 1391 | 0.1 | 100 | | | 1.4 | 7 | 0.74 |
| | K-EC-SO$_x$ | 57553 | 5 | 100 | | 191 | 1.4 | 4 | 0.85 |
| | K-EC-Oxalate | 161225 | 13 | 0.3 | | | 1.4 | 7 | 0.74 |
| K-rich | K-CN | 41740 | 3 | 35 | | | 1.8 | 17 | 0.76 |
| | K-NO$_x$ | 12078 | 1 | 100 | 3118 | | 1.8 | 1 | 0.91 |
| | K-SO$_x$ | 296512 | 25 | 9 | | 782 | 1.8 | 3 | 0.85 |
| | K-SO$_x$-Oxalate | 28754 | 2 | 100 | 4988 | | 1.8 | 2 | 0.97 |
| | K-Aluminosilicate | 3797 | 0.3 | 21 | 23 | | 2 | 2 | 0.91 |
| | K-Na | 2500 | 0.2 | 14 | | | 2 | 1 | 0.79 |
| Na-rich | Sea salt-fresh | 26175 | 2 | 68 | | | 2.2 | 6 | 1.54 |
| | Sea salt-aged | 69566 | 6 | 59 | 270 | | 2.2 | 3 | 1.81 |
| | Na-EC | 1415 | 0.1 | 100 | | | 2.2 | 3 | 0.59 |
| Amines | K-TMA | 25603 | 2 | 0 | | | 1.5 | 1 | 0.74 |
| | TMA-EC | 19688 | 2 | 0.3 | | | 1.5 | 1 | 0.78 |
| | OC-TMA | 3734 | 0.3 | 18 | | | 1.5 | 0.5 | 0.70 |
| OC-rich | OC | 13323 | 1 | 2 | 110 | | 1.8 | 2 | 0.74 |
| | K-OC-NO$_x$ | 1368 | 0.1 | 100 | | | 1.8 | 0.2 | 0.66 |
| | K-OC-SO$_x$ | 7435 | 1 | 95 | | | 1.8 | 1 | 0.64 |
| V-rich | V | 9810 | 1 | 9 | 71 | 65 | 3.1 | 2 | 0.70 |
| | EC-V | 3827 | 0.3 | 43 | | 198 | 3.1 | 1 | 0.69 |
| Fe-rich | Fe | 2199 | 0.2 | 71 | 215 | 37 | 3.6 | 0.5 | 0.88 |
| Ca-rich | Ca | 2400 | 0.2 | 82 | 352 | 14 | 2.6 | 1 | 1.08 |





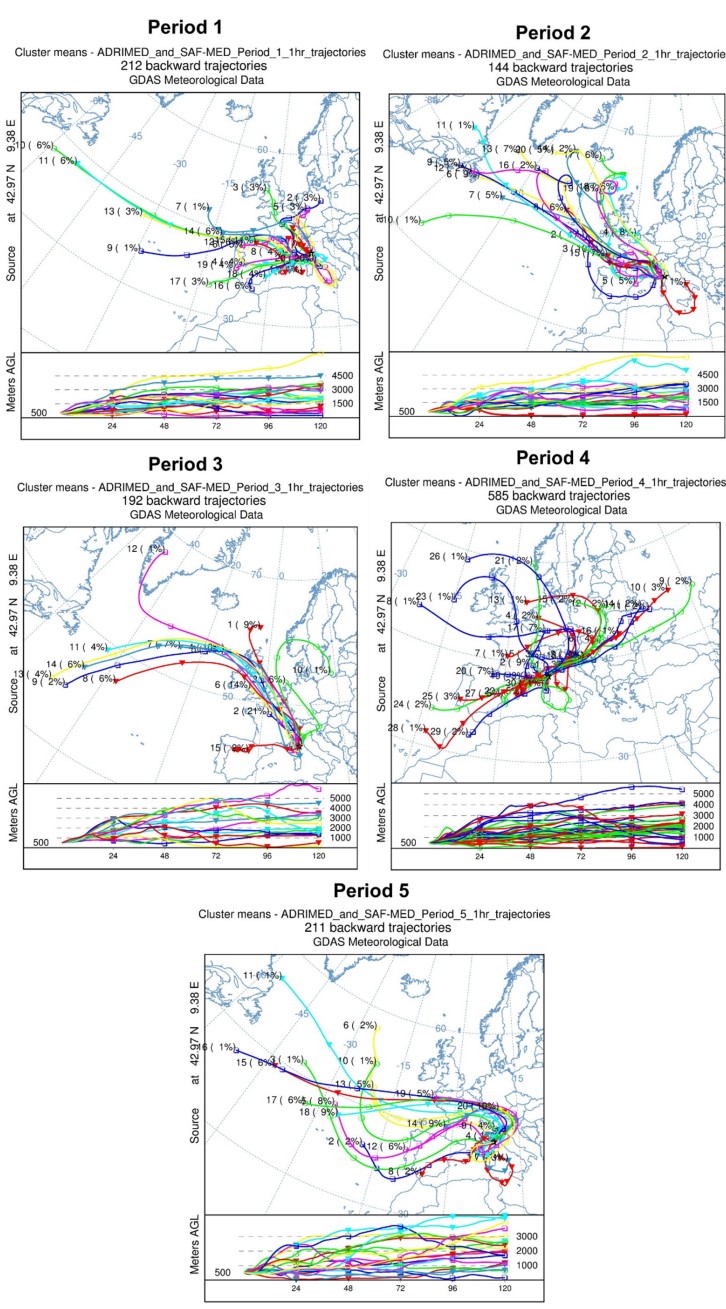

**Figure 1.** Cluster analysis of 120-hour back trajectories calculated using the HYPLSIT model, ending at Cap Corse sampling site at 500 meters above ground level, every 1 hour, for the five periods identified during ADRIMED and SAF-MED.





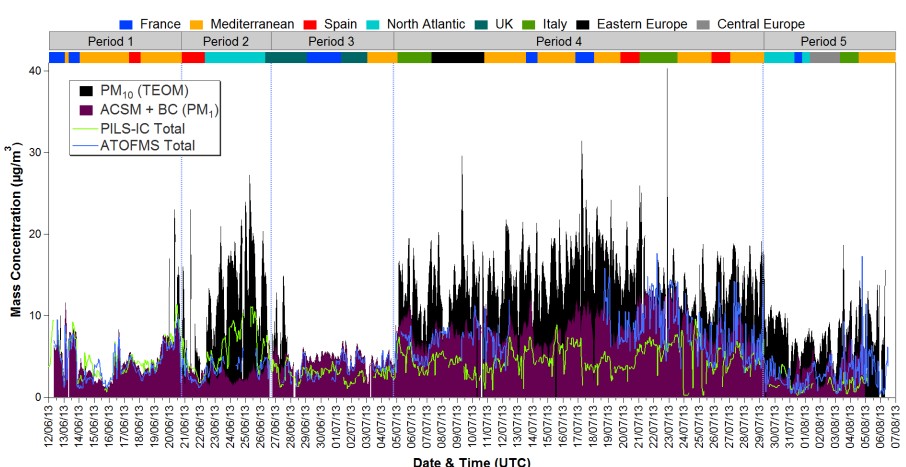

**Figure 2.** Hourly mass concentrations of $PM_{10}$, total ACSM species + BC ($PM_1$), total PILS-IC species ($PM_{10}$) and total ionised ATOFMS particles during ADRIMED and SAF-MED.





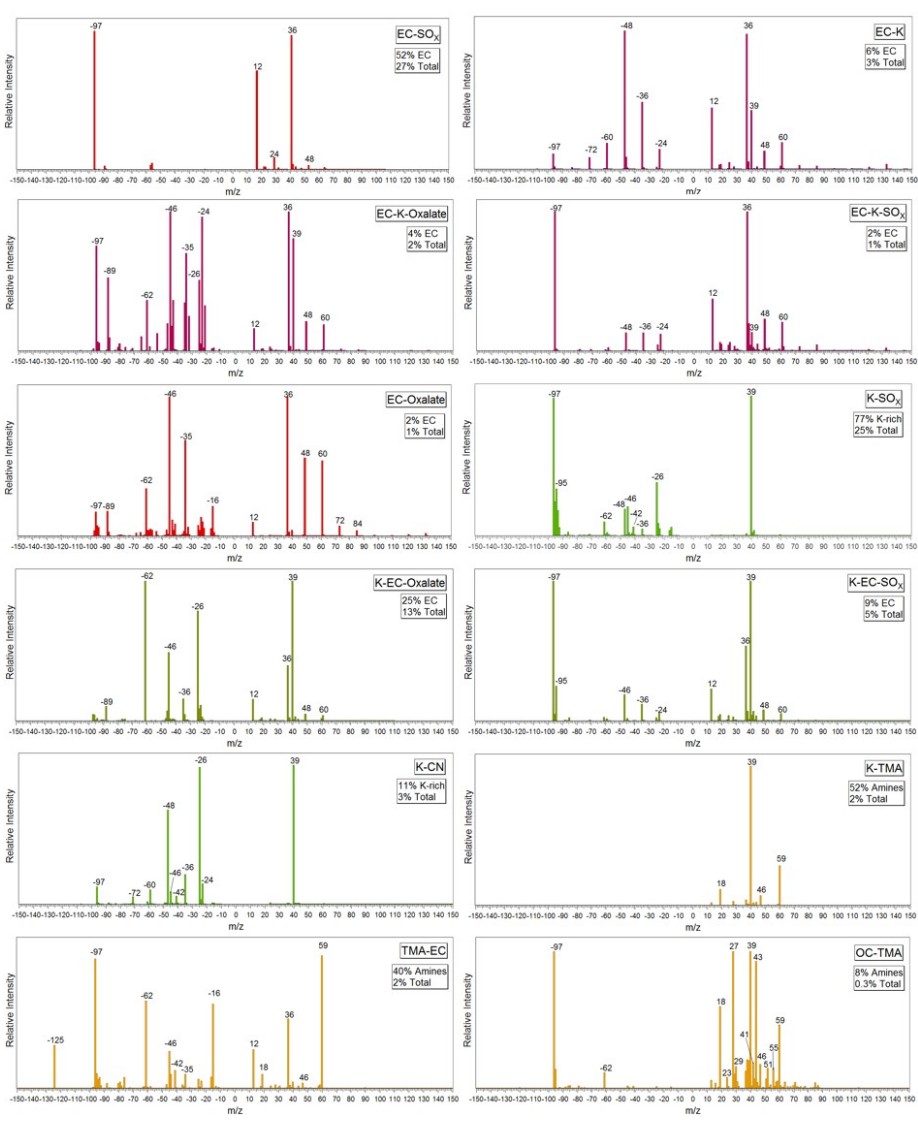

**Figure 3.** Average mass spectra of ATOFMS particle classes originating from regionally transported combustion sources during ADRIMED and SAF-MED. Percentages refer to fraction of ATOFMS category and total particle numbers.





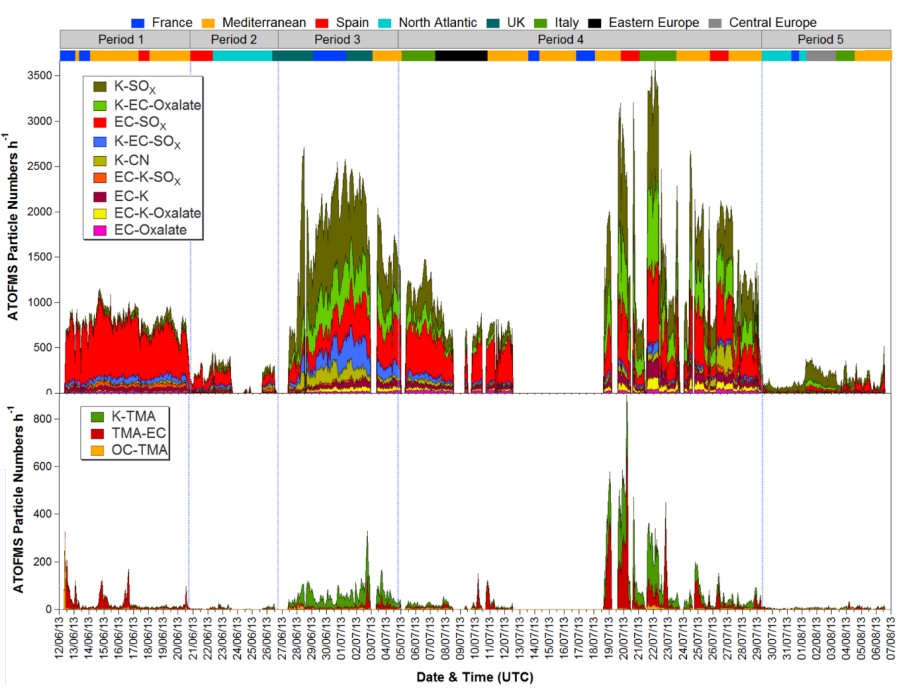

**Figure 4.** Time series (stacked) of hourly unscaled particle numbers for ATOFMS EC and major K-rich particle classes (top) and Amine classes (bottom) observed during ADRIMED and SAF-MED.





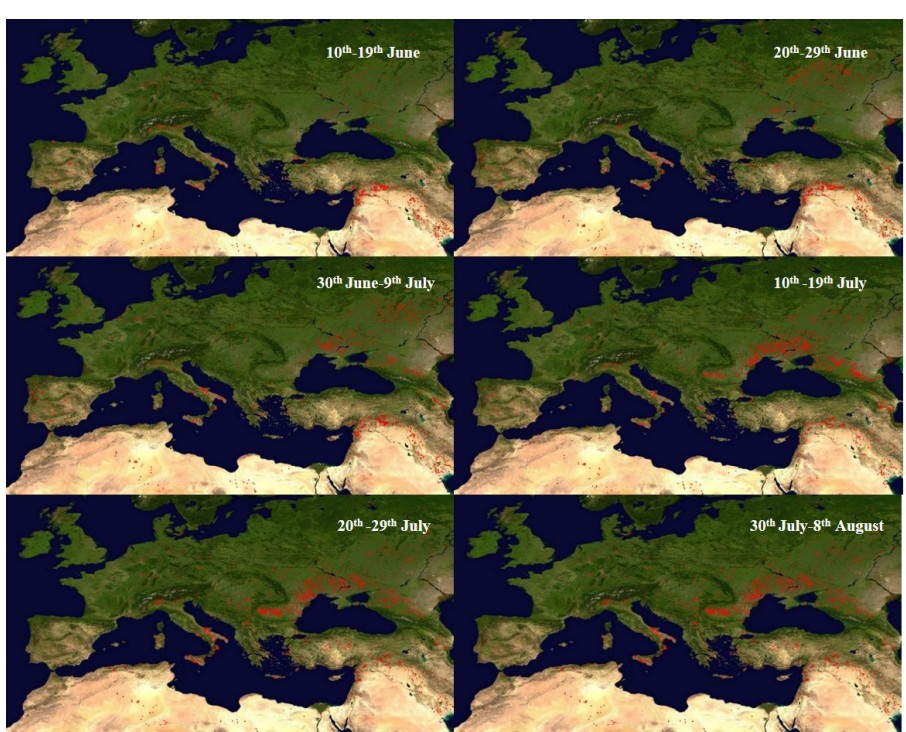

**Figure 5.** Locations of fires detected by the MODIS sensor on board the Terra and Aqua satellites over 10-day periods during ADRIMED and SAF-MED. Each red dot indicates a location where at least one fire was detected. (http://lance-modis.eosdis.nasa.gov/cgi-bin/imagery/firemaps.cgi)





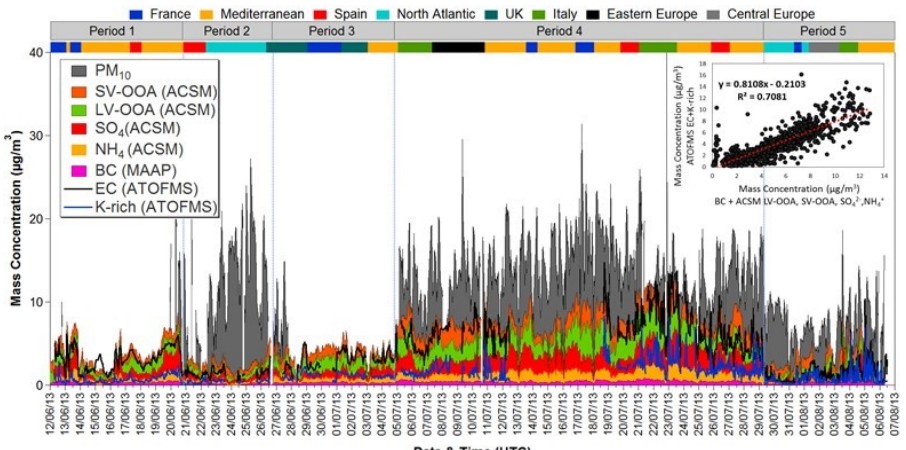

**Figure 6.** Hourly mass concentrations of PM$_{10}$, ACSM species (SO$_4^{2-}$, NH$_4^+$, organic aerosol factors SV-OOA and LV-OOA), BC and reconstructed ATOFMS EC and K-rich particles. BC and ACSM species profiles are stacked, as are both ATOFMS categories, but separately. This compares the ATOFMS mass concentrations with BC+ACSM species. Inset is a scatter plot of mass concentrations of ATOFMS EC-rich+K-rich particles compared with those for BC+SV-OOA+LV-OOA+SO$_4^{2-}$+NH$_4^+$.

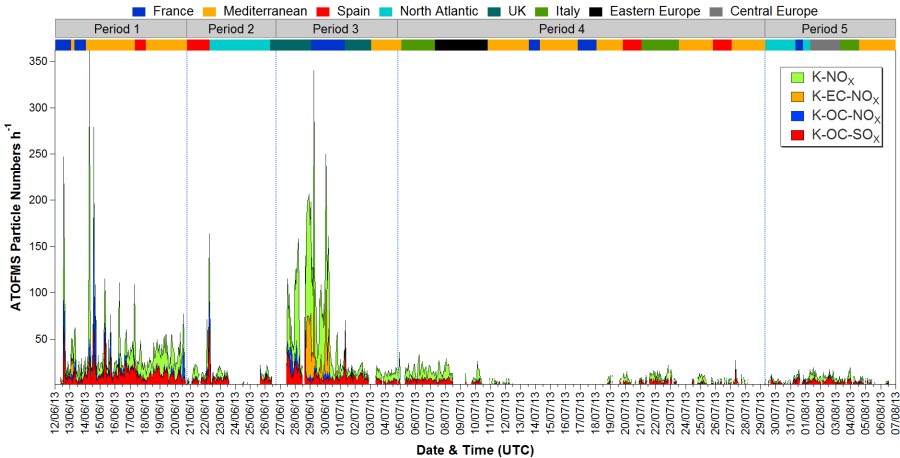

**Figure 7.** Time series of hourly unscaled particle numbers for ATOFMS particle classes associated with local biomass burning observed during ADRIMED and SAF-MED.

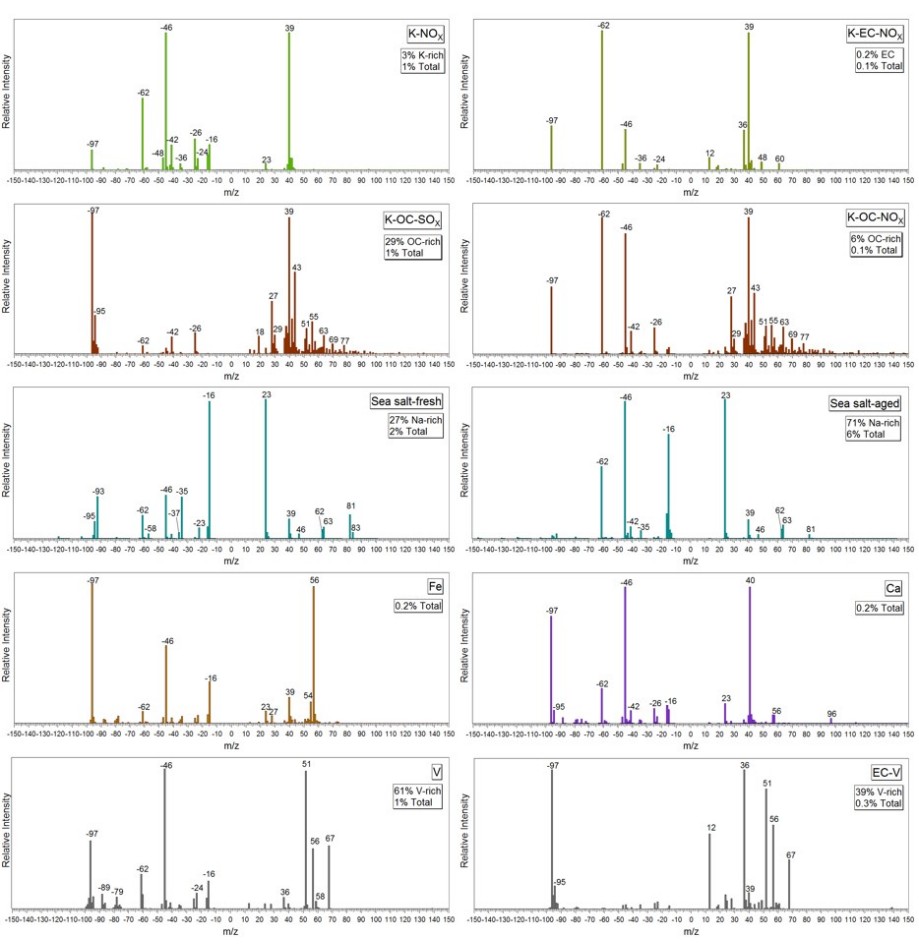

**Figure 8.** Average mass spectra for ATOFMS particle classes from local biomass burning, marine, dust and shipping sources during ADRIMED and SAF-MED.



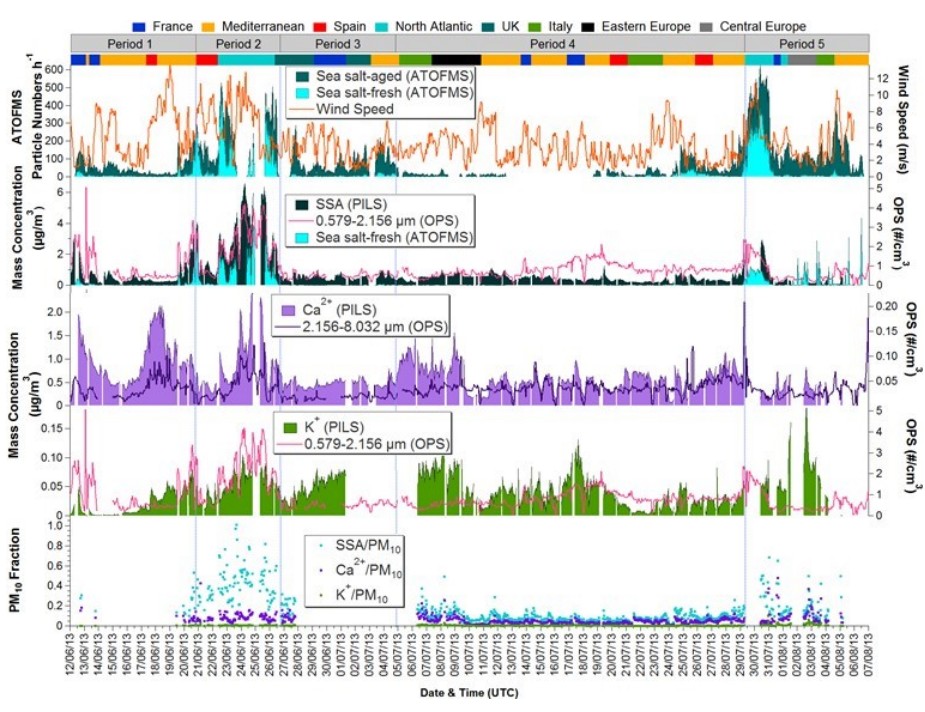

**Figure 9.** Time series of hourly ATOFMS sea salt particle numbers, OPS number concentrations, PILS-IC SSA (sea salt aerosol), Ca$^+$, K$^+$ and ATOFMS fresh sea salt mass concentrations, and PM$_{10}$ fractions observed during ADRIMED and SAF-MED.





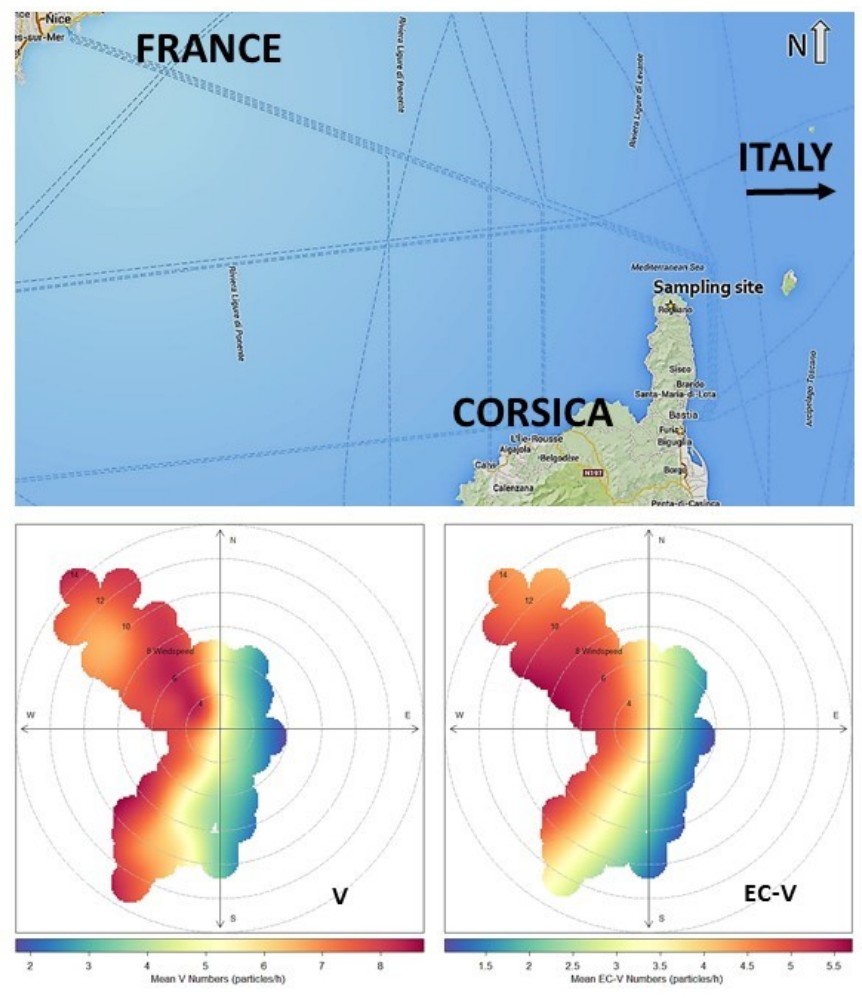

**Figure 10.** Passenger ferry lanes around the sampling site on Corsica (top) and wind speed and direction dependences (bottom) for ATOFMS V-rich particles observed during ADRIMED and SAF-MED.





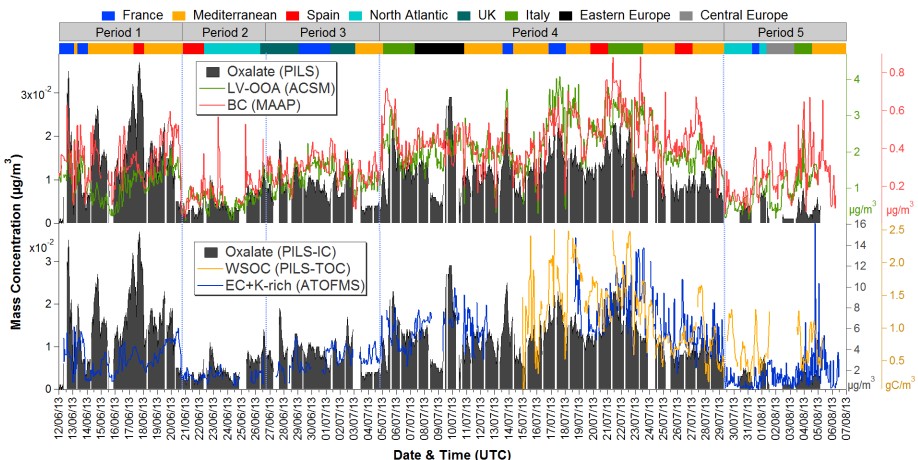

**Figure 11.** Hourly mass concentrations of the ACSM factor SV-OOA, BC (black carbon), WSOC (water soluble organic carbon) and ATOFMS EC+K-rich particles during ADRIMED and SAF-MED.

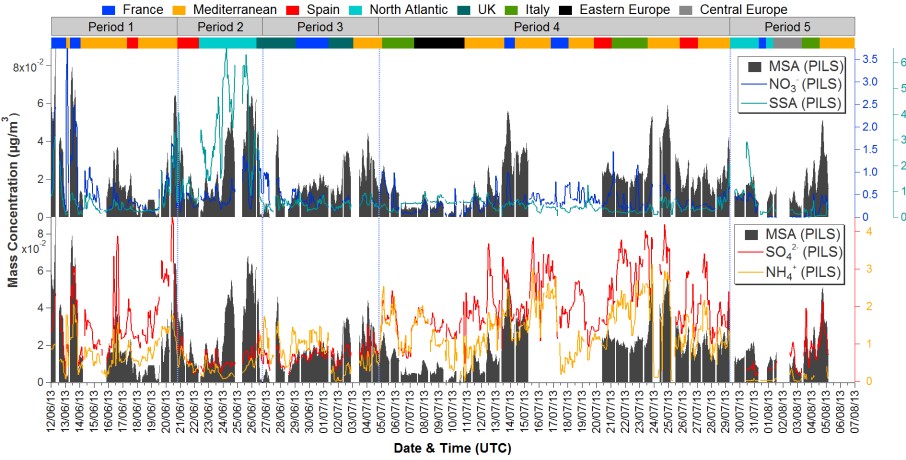

**Figure 12.** Hourly mass concentrations of PILS-IC species MSA (methanesulfonate), $NO_3^-$, SSA (sea salt aerosol), $SO_4^{2-}$ and $NH_4^+$ during ADRIMED and SAF-MED.