# Peer review of "Sources and mixing state of summertime background aerosol in the northwestern Mediterranean basin"

_Atmospheric Chemistry and Physics, 2016_

## Referee Comment (RC1) · Anonymous Referee #2 · 26 Feb 2017

An aerosol time-of-flight mass spectrometer (ATOFMS) was employed to provide real-time single particle mixing state and thereby source information for aerosols impacting the western Mediterranean basin during the ChArMEx-ADRIMED and SAF-MED campaigns in summer 2013. The ATOFMS measurements were made at a ground-based remote site on the northern tip of Corsica Island. ATOFMS particle classes were identified and grouped into 8 general categories: EC, K-rich, Na-rich, Amines, OC-rich, V-rich, Fe-rich and Ca-rich. ATOFMS reconstructed PM2.5 mass was regionally transported fossil fuel (EC-rich) and biomass burning (K-rich) particles. As the authors mentioned in their conclusion chapter, I fully agree that the identification of these sources and apportioning aerosol mass to them is a key component of future work to mitigate

their effects on the Mediterranean climate, however the authors often the term "suggesting that. . ." which sounds as some of their findings are based on hypothesis rather than robust evidences. My overall judging is that this study is of good quality and deserves publication in ACP, after treating carefully the major comments raised and simplifying the text so as to be clearer to the potential reader. Major comments: 1) As mentioned above, the authors ought to present stronger arguments to some of the interpretations of their results rather than suggesting certain possible reasons for the results obtained (see, for example in p.9 l.31; p.10 l.37-39; p.11 l.11; p.14 l. 26; p.15 l.3; p.17 l. 9 and l.27). 2) P.3 l.18: Please explain why only $O_3$ and $PM_{2.5}$ are formed in arid conditions and strong insolation. 3) P.5 l.12-14: Please explain what are the scientific consequences of the degradation in the power of the sizing lasers observed during ADRIMED and SAF-MED experiments. 4) P.6 l.30-31: Please explain along which criteria the densities were estimated for each class. 5) P.7 l.18: Please give some arguments on the selection of 500 m as release height of the back trajectories generated. Other comments: 6) P.3 l.14: The sentence beginning with the words:" The geography and regional processes. . ." is expressed in a too general manner, please elaborate. 7) P.3 l.9-10: In the context of the Asian monsoon outflow transporting pollution in the upper troposphere, across northern Africa and the Mediterranean, please refer also to Ricaud et al. (2014).

Ref: Ricaud, P. et al (2014): Impact of the Asian monsoon anticyclone on the variability of mid-to-upper tropospheric methane above the Mediterranean Basin, ATMOSPHERIC CHEMISTRY AND PHYSICS, 14, 20, 11427-11446, DOI: 10.5194/acp-14-11427-2014.

---

## Referee Comment (RC2) · Anonymous Referee #1 · 8 Mar 2017

Sources and mixing state of summertime background aerosol in the northwestern Mediterranean basin

Jovanna Arndt et al.
ACP-2016-1044

**Summary**

The manuscript documents a study about the aerosol environment in northwestern Mediterranean based on the data obtained during two intense sampling periods of ChArMEx-ADRIMED and SAF-MED campaigns in summer 2013. Aerosol properties were measured by a number of instruments, and the analysis involved primarily ATOFMS, provided significant information regarding aerosol mixing state.
By making use of statistical techniques, k-mean clustering method, analysis of positive ions and negative ions spectral shape, the more than a million particle spectra obtained by ATOFMS were reduced to small number of particle classes and source apportionment was carried by referring backward trajectory analysis and some understandings of commercial, industrial, transportation, agricultural activities in the surrounding regions.

It is a well-written and organized manuscript; it offers significant information about the aerosols and their sources affecting the NW Mediterranean. It connects aerosol measurement to future possible studies of aerosol impacts on regional climate in NW Mediterranean. I recommend it for publishing in ACP after addressing some minor comments list below.

**General comment**

1. In the paragraph (line 165-170), it mentioned the conversion of diameters, and the conversion assumed the spherical shape of the particles. Could you please provide some more information about the shape of the particles detected in the campaign? Furthermore could you provide some discussion about the impact of the results of the particles classification and the conclusion if some of the particles are not spherical?

2. In the paragraph (line 171-175), it discussed the conversion of diameters requires the density. It is not clear to me how to obtain the density values, specifically, firstly how to obtain the equation on line 175? Secondly, based on equation on line 175, do you assume one density value, like an average density values when doing the diameter conversion for all particles? It is not entirely clear. Thirdly, any assumption needed in deriving equation on line 175? Last, could you elaborate how to use to measurements from like MAAP or ACSM or any other instrument you needed in this study to obtain the density based on equation on line 175?

3. For the ATOFMS analysis, it would be great if there is a figure showing the schematic how the 1.2 million single particle mass spectra obtained by ATOFMS during the sampling period being reduced to 80 clusters, then 27 classes and furthermore linked to source apportionment and background trajectory analyses. It would enhance the readers'

understanding and help readers quickly get across the key message of the manuscript. I suggest the authors add such schematic diagram.

---

## Author Comment (AC1) · 28 Apr 2017

We would like to thank the reviewer for their valuable comments and suggestions. A fully formatted pdf version of our responses is also attached.

RC1 (Anonymous Referee #2):

An aerosol time-of-flight mass spectrometer (ATOFMS) was employed to provide real-time single particle mixing state and thereby source information for aerosols impacting the western Mediterranean basin during the ChArMEx-ADRIMED and SAF-MED campaigns in summer 2013. The ATOFMS measurements were made at a ground-based remote site on the northern tip of Corsica Island. ATOFMS particle classes were identified and grouped into 8 general categories: EC, K-rich, Na-rich, Amines, OC-rich, V-rich, Fe-rich and Ca-rich. ATOFMS reconstructed PM2.5 mass was regionally transported fossil fuel (EC-rich) and biomass burning (K-rich) particles. As the authors mentioned in their conclusion chapter, I fully agree that the identification of these sources and apportioning aerosol mass to them is a key component of future work to mitigate their effects on the Mediterranean climate, however the authors often the term "suggesting that. . ." which sounds as some of their findings are based on hypothesis rather than robust evidences. My overall judging is that this study is of good quality and deserves publication in ACP, after treating carefully the major comments raised and simplifying the text so as to be clearer to the potential reader.

Major comments: As mentioned above, the authors ought to present stronger arguments to some of the interpretations of their results rather than suggesting certain possible reasons for the results obtained (see, for example in p.9 l.31; p.10 l.37-39; p.11 l.11; p.14 l. 26; p.15 l.3; p.17 l. 9 and l.27).

Response: Several instances of "suggesting" have been rewritten in order to better reflect our confidence in the results and the conclusions drawn from them, as follows:

p.5 l.138: "The particle class labelling scheme used herein is regularly used in the literature (Ault et al., 2010; Dall'Osto and Harrison, 2006; Pratt and Prather, 2012; Spencer and Prather, 2006) and indicates either the probable source (e.g. sea salt) or the dominant species in the positive ion mass spectra (e.g. K, EC, Fe etc.), with the order of the ions signifying their relative mass spectral intensities."

p.8 l.258: "EC-SOX and EC-Oxalate did not contain detectable 39K+, a sign that they most likely arise from fossil fuel combustion (oil burning or traffic) and not biomass burning."

p.9 l.282: "EC fragments were found in all K-rich negative ion mass spectra, typical of a biomass combustion source."

P.9 l.288: "The K-TMA class was dominated by 39K+, a typical marker of biomass burning, while EC-TMA particles produced 12,36C3+ signals, markers for fossil fuel combustion origins. OC-TMA particles are characterised by strong 39,41K+ and OC (27C2H3+), another typical marker for biomass burning sources, and oxidised OC (43C2H3O+) signals, indicative of atmospheric processing during transport to the site."

p. 11 l.341: "No association between any of the Amine classes and local relative humidity was found in this case. This shows that this effect is not relevant close to the receptor site but may have played a role close to the point of emission or during transport of the Amine particles to the site."

P.11 l.359: "Individual particle classes did not produce stronger correlations, demonstrating that no single class was an important contributor of PM2.5 composition."

p.12 l.398: "K-OC-SOX particles also exhibited a small MSA (95(CH)3SO3-) signal, demonstrating at least some mixing with marine biogenic emissions prior to detection."

p.15 l.509: "In our study, oxalate was found in ca. 9600 particles, i.e. 0.8% of the total particles ionised. The mixing states derived from this query are similar to those produced from general clustering and are varied. Poor ionisation efficiency did therefore not prevent oxalate from being detected in certain types of particles."

P.15 l.485: "EC-V and V particle numbers consistently featured a mode around 740 nm Da, a sign that the observed particles were chemically processed to some degree during transport."

p.16 l.522: "The queried oxalate particle numbers were considered representative of the whole ATOFMS dataset; biomass burning emissions therefore play a large role in the fate of particle phase oxalate in the western Mediterranean."

p.16 l.525: "The ATOFMS querying approach indicated a prevalence of oxalate mixed with K-rich particles, however correlations between PILS-IC oxalate and K-rich mass were poor. Stronger correlations were found with EC-rich and K-rich mass concentrations combined (R2=0.55); either more EC particles actually contained oxalate than was detected or another particle type was transported in the same air mass but was not detected by the ATOFMS."

p.19 l.538: "Previous studies of amine-containing particles found a strong dependence on relative humidity. This was not the case during these two campaigns, showing that these particles were not formed locally."

P.3 l.18: Please explain why only O3 and PM2.5 are formed in arid conditions and strong insolation.

Response: We will amend this line to read: "Arid conditions result in less wet deposition of particles and increase aerosol lifetimes, while high solar radiation and photochemical conversion rates significantly enhance air pollution in the form of O3."

P.5 l.12-14: Please explain what are the scientific consequences of the degradation in the power of the sizing lasers observed during ADRIMED and SAF-MED experiments.

Response: We will add the following lines after P.5 l.12-14 to address this comment: "This is a more limited size range compared to the normal 100-3000 nm operating range of the instrument. It is relevant for interpretation of unscaled size distributions of particle classes, particularly combustion-related particles. Many of these are less than 500 nm and as such would not have been detected by the instrument. These missing particles would in turn affect the reconstructed ATOFMS mass concentrations, as only particles larger than 300 and 500 nm would have been used for the data analysis. However, as detailed in the discussions below, reconstructed mass ATOFMS concentrations agreed well with co-located quantitative measurements and as such this missing particle mass did not invalidate the use of the analysis."

P.6 l.30-31: Please explain along which criteria the densities were estimated for each class.

Response: Apologies for the unnecessary confusion – the following text will be added

after the equation on line 175:

"Where m is the average mass of BC and ACSM species. 1.5, 1.2, 1.52 and 1.75 (Allan et al., 2003) are material densities for BC, organic aerosol (Org), non-sea salt Cl-, SO42-, NO3- and NH4+ respectively. An average estimated density of 1.4 g/cm3 was observed for bulk aerosol for the ADRIMED and SAF-MED campaigns. From the density calculation it is clear that neither metal-rich nor sea salt particles are taken into account. From the PILS-IC (PM10) it was clear that sea salt particles constituted a significant fraction of PM10 aerosol (6% overall, 40-50% during the major sea salt event). The average density was therefore expected to be larger, thus a density of 1.7 (Reinard et al., 2007) was used to convert the diameters. Mass concentrations can be obtained from the scaled number concentrations by (Reinard et al., 2007):

(Equation can be found in the attached pdf)

A precise transformation of number to mass concentration requires knowledge of $\chi$ and p for each particle class. As discussed above, $\chi$ is assumed to be 1. The use of a single density, p, for ATOFMS scaling has previously resulted in satisfactory PM mass reconstruction when compared to other quantitative measurements (Healy et al., 2012, 2013; Qin et al., 2006). However, a single density assumption is known to be incorrect due to differing particle compositions (Maricq and Xu, 2004; Spencer and Prather, 2006). Different particle classes will exhibit different particle densities. A range of densities was therefore used to calculate mass concentrations for each particle class, which can be found in Table 2. The class densities were estimated from the bulk densities of the chemical components indicated in the mass spectra as described by Bein et al. (2006) and Reinard et al. (2007)."

P.7 l.18: Please give some arguments on the selection of 500 m as release height of the back trajectories generated.

Response: We will amend this line to read: "120-hour back trajectories ending 500 m above ground level (AGL) at Ersa (Corsica), approximately equivalent to the site

altitude, were calculated for each hour between 12th June and 7th August 2013 (total: 1344 trajectories).

P.3 l.14: The sentence beginning with the words:" The geography and regional processes. . ." is expressed in a too general manner, please elaborate.

Response: We will add the following content, and add the new references to the bibliography: "The geography and regional meteorological processes in the western Mediterranean also favour the accumulation and ageing of polluted air masses (Gangoiti, 2001; Lelieveld et al., 2002; Millán et al., 2000, 2002; Millán and Salvador, 1997; Rodríguez et al., 2002; Salvador et al., 1999; Soriano et al., 2001). The Iberian Peninsula on the western coast of the basin, the Alps to the north, the Apennines and Balkans to the east and the Atlas Mountains to the south act as physical barriers between the frontal weather systems of northern Europe and the Sahara, and the Inter Tropical Fronts in the south. In summer the Mediterranean meteorological situation is characterised by two high-pressure ridges – one over central Europe and one over the western Mediterranean basin – and a deep trough extending from the Persian Gulf to the eastern Mediterranean basin. These systems lead to low winds, persistent clear-sky conditions, high solar irradiation, anticyclonic subsidence and stratification (Anagnostopoulou et al., 2014; Doche et al., 2014; Tyrlis and Lelieveld, 2013)."

Anagnostopoulou, C., Zanis, P., Katragkou, E., Tegoulias, I. & Tolika, K. (2014). Recent past and future patterns of the Etesian winds based on regional scale climate model simulations. Climate Dynamics. 42 (7-8). p.pp. 1819–1836. Doche, C., Dufour, G., Foret, G., Eremenko, M., Cuesta, J., Beekmann, M. & Kalabokas, P. (2014). Summertime tropospheric-ozone variability over the Mediterranean basin observed with IASI. Atmospheric Chemistry and Physics. 14 (19). p.pp. 10589–10600. Tyrlis, E. & Lelieveld, J. (2013). Climatology and Dynamics of the Summer Etesian Winds over the Eastern Mediterranean*. Journal of the Atmospheric Sciences. 70 (11). p.pp. 3374–3396.

P.3 l.9-10: In the context of the Asian monsoon outflow transporting pollution in the upper troposphere, across northern Africa and the Mediterranean, please refer also to Ricaud et al. (2014).

Response: Agreed and we will add this reference in the final manuscript.

Please also note the supplement to this comment:
http://www.atmos-chem-phys-discuss.net/acp-2016-1044/acp-2016-1044-AC1-supplement.pdf

---

## Author Comment (AC2) · 28 Apr 2017

We would like to thank the reviewer for their valuable comments and suggestions. A fully formatted pdf version of our responses is also attached.

RC2 (Anonymous Referee #1): The manuscript documents a study about the aerosol environment in northwestern Mediterranean based on the data obtained during two intense sampling periods of ChArMEx-ADRIMED and SAF-MED campaigns in summer 2013. Aerosol properties were measured by a number of instruments, and the analysis involved primarily ATOFMS, provided significant information regarding aerosol mixing state. By making use of statistical techniques, k-mean clustering method, analysis of positive ions and negative ions spectral shape, the more than a million particle

spectra obtained by ATOFMS were reduced to small number of particle classes and source apportionment was carried by referring backward trajectory analysis and some understandings of commercial, industrial, transportation, agricultural activities in the surrounding regions. It is a well-written and organized manuscript; it offers significant information about the aerosols and their sources affecting the NW Mediterranean. It connects aerosol measurement to future possible studies of aerosol impacts on regional climate in NW Mediterranean. I recommend it for publishing in ACP after addressing some minor comments list below.

In the paragraph (line 165-170), it mentioned the conversion of diameters, and the conversion assumed the spherical shape of the particles. Could you please provide some more information about the shape of the particles detected in the campaign? Furthermore could you provide some discussion about the impact of the results of the particles classification and the conclusion if some of the particles are not spherical?

Response: Unfortunately the ATOFMS is not capable of measuring particle shape. Any potential shape information would need to have been attributable to specific particle classes; again unfortunately this was not available. It is known that not all particle classes are spherical; for example elemental carbon typically takes the form of soot agglomerates, sea salt is non-spherical. However, atmospheric processing of particles, where they become coated with secondary species such as ammonium nitrate or ammonium sulfate typically produces a more spherical particle. This was expected to be the case for the majority of the particles observed at this site, after undergoing regional transport. The result of these factors was the use of a single shape value and assumption of sphericity for all particles in converting from aerodynamic diameter to volume equivalent diameter.

In the paragraph (line 171-175), it discussed the conversion of diameters requires the density. It is not clear to me how to obtain the density values, specifically, firstly how to obtain the equation on line 175? Secondly, based on equation on line 175, do you assume one density value, like an average density values when doing the diameter

conversion for all particles? It is not entirely clear. Thirdly, any assumption needed in deriving equation on line 175? Last, could you elaborate how to use to measurements from like MAAP or ACSM or any other instrument you needed in this study to obtain the density based on equation on line 175?

Response: Apologies for the unnecessary confusion – the following text, which addresses each of the points raised by the reviewer here, will be added after the equation on line 175: "Where m is the average mass of BC and ACSM species. 1.5, 1.2, 1.52 and 1.75 (Allan et al., 2003) are material densities for BC, organic aerosol (Org), non-sea salt Cl-, $SO_4^{2-}$, $NO_3^-$ and $NH_4^+$ respectively. An average estimated density of 1.4 g/cm3 was observed for bulk aerosol for the ADRIMED and SAF-MED campaigns. From the density calculation it is clear that neither metal-rich nor sea salt particles are taken into account. From the PILS-IC (PM10) it was clear that sea salt particles constituted a significant fraction of PM10 aerosol (6% overall, 40-50% during the major sea salt event). The average density was therefore expected to be larger, thus a density of 1.7 (Reinard et al., 2007) was used to convert the diameters. Mass concentrations can be obtained from the scaled number concentrations by (Reinard et al., 2007):

(Equation can be found in attached pdf)

A precise transformation of number to mass concentration requires knowledge of $\chi$ and p for each particle class. As discussed above, $\chi$ is assumed to be 1. The use of a single density, p, for ATOFMS scaling has previously resulted in satisfactory PM mass reconstruction when compared to other quantitative measurements (Healy et al., 2012, 2013; Qin et al., 2006). However, a single density assumption is known to be incorrect due to differing particle compositions (Maricq and Xu, 2004; Spencer and Prather, 2006). Different particle classes will exhibit different particle densities. A range of densities was therefore used to calculate mass concentrations for each particle class, which can be found in Error! Reference source not found.. The class densities were estimated from the bulk densities of the chemical components indicated in the mass spectra as described by Bein et al. (2006) and Reinard et al. (2007)."

For the ATOFMS analysis, it would be great if there is a figure showing the schematic how the 1.2 million single particle mass spectra obtained by ATOFMS during the sampling period being reduced to 80 clusters, then 27 classes and furthermore linked to source apportionment and background trajectory analyses. It would enhance the readers' understanding and help readers quickly get across the key message of the manuscript. I suggest the authors add such schematic diagram.

Response: We agree that this would improve the reader's understanding of the numerous steps in the data analysis employed in this manuscript and will include a schematic diagram of the process in the supporting information.

Please also note the supplement to this comment:
http://www.atmos-chem-phys-discuss.net/acp-2016-1044/acp-2016-1044-AC2-supplement.pdf

[Figure]

**Fig. 1.**

**Supplement:**

We would like to thank the reviewer for their valuable comments and suggestions.

RC2 (Anonymous Referee #1):

The manuscript documents a study about the aerosol environment in northwestern Mediterranean based on the data obtained during two intense sampling periods of ChArMEx-ADRIMED and SAF-MED campaigns in summer 2013. Aerosol properties were measured by a number of instruments, and the analysis involved primarily ATOFMS, provided significant information regarding aerosol mixing state. By making use of statistical techniques, k-mean clustering method, analysis of positive ions and negative ions spectral shape, the more than a million particle spectra obtained by ATOFMS were reduced to small number of particle classes and source apportionment was carried by referring backward trajectory analysis and some understandings of commercial, industrial, transportation, agricultural activities in the surrounding regions. It is a well-written and organized manuscript; it offers significant information about the aerosols and their sources affecting the NW Mediterranean. It connects aerosol measurement to future possible studies of aerosol impacts on regional climate in NW Mediterranean. I recommend it for publishing in ACP after addressing some minor comments list below.

1) *In the paragraph (line 165-170), it mentioned the conversion of diameters, and the conversion assumed the spherical shape of the particles. Could you please provide some more information about the shape of the particles detected in the campaign? Furthermore could you provide some discussion about the impact of the results of the particles classification and the conclusion if some of the particles are not spherical?*

**Response:**

Unfortunately the ATOFMS is not capable of measuring particle shape. Any potential shape information would need to have been attributable to specific particle classes; again unfortunately this was not available. It is known that not all particle classes are spherical; for example elemental carbon typically takes the form of soot agglomerates, sea salt is non-spherical. However, atmospheric processing of particles, where they become coated with secondary species such as ammonium nitrate or ammonium sulfate typically produces a more spherical particle. This was expected to be the case for the majority of the particles observed at this site, after undergoing regional transport. The result of these factors was the use of a single shape value and assumption of sphericity for all particles in converting from aerodynamic diameter to volume equivalent diameter.

2) *In the paragraph (line 171-175), it discussed the conversion of diameters requires the density. It is not clear to me how to obtain the density values, specifically, firstly how to obtain the equation on line 175? Secondly, based on equation on line 175, do you assume one density value, like an average density values when doing the diameter conversion for all particles? It is not entirely clear. Thirdly, any assumption needed in deriving equation on line 175? Last, could you elaborate how to use to measurements from like MAAP or ACSM or any*

*other instrument you needed in this study to obtain the density based on equation on line 175?*

**Response:**

Apologies for the unnecessary confusion – the following text, which addresses each of the points raised by the reviewer here, will be added after the equation on line 175:

"Where $m$ is the average mass of BC and ACSM species. 1.5, 1.2, 1.52 and 1.75 (Allan et al., 2003) are material densities for BC, organic aerosol (Org), non-sea salt $Cl^-$, $SO_4^{2-}$, $NO_3^-$ and $NH_4^+$ respectively. An average estimated density of 1.4 $g/cm^3$ was observed for bulk aerosol for the ADRIMED and SAF-MED campaigns. From the density calculation it is clear that neither metal-rich nor sea salt particles are taken into account. From the PILS-IC ($PM_{10}$) it was clear that sea salt particles constituted a significant fraction of $PM_{10}$ aerosol (6% overall, 40-50% during the major sea salt event). The average density was therefore expected to be larger, thus a density of 1.7 (Reinard et al., 2007) was used to convert the diameters.

Mass concentrations can be obtained from the scaled number concentrations by (Reinard et al., 2007):

$$m = \frac{\pi}{6} \rho_p d_{ve}^3$$

A precise transformation of number to mass concentration requires knowledge of $\chi$ and $\rho_p$ for each particle class. As discussed above, $\chi$ is assumed to be 1. The use of a single density, $\rho_p$, for ATOFMS scaling has previously resulted in satisfactory PM mass reconstruction when compared to other quantitative measurements (Healy et al., 2012, 2013; Qin et al., 2006). However, a single density assumption is known to be incorrect due to differing particle compositions (Maricq and Xu, 2004; Spencer and Prather, 2006). Different particle classes will exhibit different particle densities. A range of densities was therefore used to calculate mass concentrations for each particle class, which can be found in **Error! Reference source not found.**. The class densities were estimated from the bulk densities of the chemical components indicated in the mass spectra as described by Bein et al. (2006) and Reinard et al. (2007)."

3) *For the ATOFMS analysis, it would be great if there is a figure showing the schematic how the 1.2 million single particle mass spectra obtained by ATOFMS during the sampling period being reduced to 80 clusters, then 27 classes and furthermore linked to source apportionment and background trajectory analyses. It would enhance the readers' understanding and help readers quickly get across the key message of the manuscript. I suggest the authors add such schematic diagram.*

**Response:** We agree that this would improve the reader's understanding of the numerous steps in the data analysis employed in this manuscript and will include the below schematic diagram of the process in the supporting information.

Thousands of single particle mass spectra

[Figure]

*K*-means algorithm

Groups similar mass spectra together

[Figure]

Particle types

Similar mass spectra, size and temporal profiles

Nomenclature based on dominant ions (usually positive e.g. K) or likely source (e.g. sea salt)

[Figure]

Scaling factors

Single particle numbers scaled up to numbers from particle counting instruments (SMPS/OPS)

Requires conversion of aerodynamic diameter to volume equivalent diameter so size bins from each instrument can be directly compared

Conversion requires selection of density value → single value = average density of ambient aerosol → applied to all particles

Reconstructed mass concentrations

Calculate volume & then mass of scaled up particle numbers

Requires selection of density value (different ones used for different particle classes) & particles assumed spherical

*Figure 1. Schematic overview of ATOFMS data analysis.*